# Generative deep learning enables the discovery of a potent and selective RIPK1 inhibitor

Yueshan Li [1,3], Liting Zhang [1,3], Yifei Wang [1,3], Jun Zou [1,3], Ruicheng Yang[1], Xinling Luo[2], Chengyong Wu[1], Wei Yang[1], Chenyu Tian[1], Haixing Xu[1], Falu Wang[1], Xin Yang[1], Linli Li[2] & Shengyong Yang [1] ✉

The retrieval of hit/lead compounds with novel scaffolds during early drug development is an important but challenging task. Various generative models have been proposed to create drug-like molecules. However, the capacity of these generative models to design wet-lab-validated and target-specific molecules with novel scaffolds has hardly been verified. We herein propose a generative deep learning (GDL) model, a distribution-learning conditional recurrent neural network (cRNN), to generate tailor-made virtual compound libraries for given biological targets. The GDL model is then applied to RIPK1. Virtual screening against the generated tailor-made compound library and subsequent bioactivity evaluation lead to the discovery of a potent and selective RIPK1 inhibitor with a previously unreported scaffold, RI-962. This compound displays potent in vitro activity in protecting cells from necroptosis, and good in vivo efficacy in two inflammatory models. Collectively, the findings prove the capacity of our GDL model in generating hit/lead compounds with unreported scaffolds, highlighting a great potential of deep learning in drug discovery.

Identifying new starting active compounds that are substantially different in chemical structure from those already on the market or in development is a crucial step in the early stage of drug development. This task is mainly accomplished by high-throughput screening, either physically or virtually, against sets of known chemical libraries. However, due to the limited structural diversity in existing chemical libraries as well as repeated screening by various companies and institutes, it is becoming more and more difficult to retrieve active compounds with new scaffolds and establish intellectual property. De novo molecular design that computationally generates new molecules with desired properties has been proposed as a solution to this problem[1–3]. Traditional de novo molecular design methods, which include structure-based[4–6], ligand-based[7,8], and pharmacophore-based methods[9,10], involve a relatively manual process that requires an experienced designer and explicit design rules. These methods are also predominately fragment based, and the quality and diversity of the generated molecules strongly depend on the fragment library and the algorithm used for fragment assembly[1].

Recently, generative deep learning (GDL) has emerged as a promising approach for de novo molecular design[3,11], where deep neural networks are employed as generative models. This approach is a completely data-driven de novo molecular design strategy without the need for explicit design rules, which can also avoid the fragment issue mentioned above. It has attracted much attention with several GDL models having been established to generate molecules, including recurrent neural network (RNN)-based[12,13], variational autoencoder (VAE)-based[14], generative adversarial network (GAN)-based[15], graph convolution network (GCN)-based[16], and transformer-based models[17].

[1]State Key Laboratory of Biotherapy and Cancer Center, West China Hospital, Sichuan University, 610041 Chengdu, Sichuan, China. [2]Key Laboratory of Drug Targeting and Drug Delivery System of Ministry of Education, West China School of Pharmacy, Sichuan University, 610041 Chengdu, Sichuan, China. [3]These authors contributed equally: Yueshan Li, Liting Zhang, Yifei Wang, Jun Zou. ✉e-mail: yangsy@scu.edu.cn

**Fig. 1 | Chemical structures of representative RIPK1 inhibitors with different scaffolds.**

Detailed description and/or comparison of these models can be found in several recent reviews[11,18,19]. Among these models, the RNN-based models are the most widely used ones, whose architectures are borrowed from the natural language processing (NLP) field with molecules being represented by a sequence of tokens, such as the simplified molecular input line entry systems (SMILES)[20]. Owing to the mature theory system of RNN, several RNN-based GDL models proposed recently produced impressive results in generating new molecules. For example, Segler et al.[12] iteratively fine-tuned a stacked RNN to generate target-focused libraries and successfully reproduced active compounds from a hold-out test set. Moret et al.[13] utilized RNN to develop a chemical language model (CLM) that enabled the discovery of new molecular entities in a low data regime. Gómez-Bombarelli et al.[14] implemented a VAE model with RNN as a decoder, which could learn to generate novel compounds with high fidelity. Kotsias et al.[21] proposed a conditional RNN (cRNN) model, in which additional molecular descriptors or fingerprints were incorporated into the RNN initial memory state to guide the subsequent generative process.

Although many GDL models including RNN-based ones showed good performance in generating molecules, a majority of them are designed to generate the best possible molecules to satisfy a pre-defined goal (goal-directed)[22]. These goal-directed models are strongly dependent on the goal functions, which may lead to the generation of molecules that are numerically superior but not practically useful[18,23]. Besides, despite that most GDL models have been demonstrated to be effective theoretically, very few have been validated by wet-lab experiments[11]. Furthermore, chemical structures of molecules generated by these models are more or less similar to those of known active compounds against the same target. To address these issues, we here propose a GDL model based on a distribution-learning cRNN[21,22], which avoids the specification of goal function and can generate new molecules following the same chemical distribution as training set molecules. Our model incorporates transfer learning[13,24], regularization enhancement[25,26], and sampling enhancement[14,27] to enable the generation of molecules with previously unreported and diverse chemical scaffolds. This model was then applied to the discovery of receptor-interacting protein kinase 1 (RIPK1) inhibitors followed by comprehensive in vitro and in vivo validations.

RIPK1 is a serine/threonine protein kinase that participates in a variety of signaling pathways involved in cell survival[28]. In particular, RIPK1 is a key regulator of programmed cell necrosis (necroptosis)[28,29], which is closely related to the occurrence and development of various inflammatory and immune diseases[30]. Mechanically, when necroptosis is triggered by stimuli such as the tumor necrosis factor family of cytokines, RIPK1 will firstly be activated. The activated RIPK1 then associates with and phosphorylates its downstream protein RIPK3, which subsequently recruits and phosphorylates the pseudokinase mixed-lineage kinase domain like (MLKL)[31,32]. The phosphorylated MLKLs form oligomers and translocate to the cytomembrane to execute necroptosis[33]. Owing to the central role of RIPK1 in necroptosis, it is considered a promising target for treating necroptosis-related diseases[30,34]. A number of RIPK1 inhibitors have been reported and five are currently under clinical trials (phase I or II) for the treatment of nervous system diseases and/or inflammatory diseases, including DNL788 (Denali; NCT05237284), DNL758 (Denali; NCT04781816), GFH312 (Genfleet; NCT04676711), SIR1−365 (Sironax; trial registered on ANZCTR: ACTRN12621000745842p) and R-552 (Rigel and Lilly; NCT05222399). Among them, only chemical structures of DNL758 and SIR1−365 are disclosed at this moment. In this study, we collected compounds with activity against RIPK1 from various publications and patents and obtained a total of 1030 compounds (Supplementary Table 1). Figure 1 shows representative compounds with different scaffold types. However, most of these reported RIPK1 inhibitors are not suitable for clinical studies due to low potency and/or poor kinase selectivity. Therefore, more RIPK1 inhibitors with previously unreported scaffolds and better potential as drug candidates should be discovered.

In this work, we present a GDL model based on a distribution-learning cRNN and then apply this model to the discovery of RIPK1 inhibitors. The rest of this article is organized as follows. We first introduce the proposed GDL model, followed by applying this model to generate a tailor-made virtual compound library targeting RIPK1 and virtual screening against this library. We next describe the retrieval of a potent and selective RIPK1 inhibitor (RI-962). The X-ray crystal structure of RIPK1 in complex with RI-962 is then illustrated. Subsequently we present the in vitro and in vivo effects of RI-962 as well as its pharmacokinetic characteristics and safety evaluation. Last are the discussion and a detailed description of the methods used.

## Results

### Establishment of the GDL model

The proposed GDL model is based on a distribution-learning cRNN architecture with the long short-term memory (LSTM) algorithm used[35]; LSTM is an advanced version of RNN that is for tackling the vanishing gradients problem. Different from traditional generative RNN models, the cRNN architecture provides an explicit initial state vector to guide the molecular generation toward a focused chemical domain, which balances the output specificity between unbiased RNN

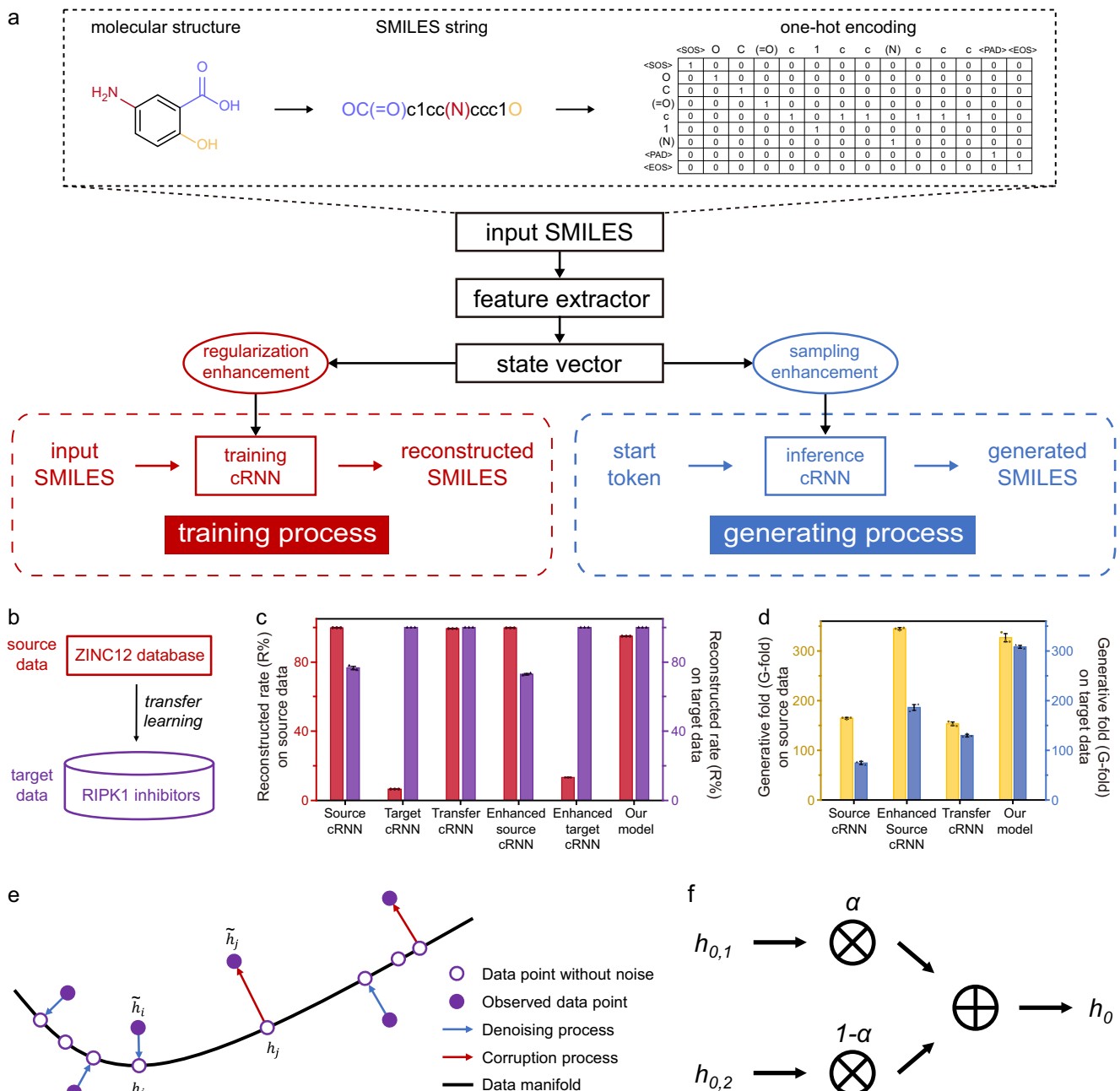

**Fig. 2 | Establishment and performance of the cRNN-based generative model.**
**a** The diagram of cRNN-based generative model. **b** Schematic of transfer learning. Molecules from the ZINC12 database (~16 million molecules) and known RIPK1 inhibitors (1030 molecules) are used as the source data and the target data, respectively. **c**, **d** The reconstruction performance (**c**) and the generation performance (**d**) of different models ($n = 3$). All results are shown as mean ± standard deviation. Source cRNN: training on the source data; target cRNN: training on the target data; transfer cRNN: training with transfer learning on the source and the target data; enhanced source cRNN: training with regularization enhancement on the source data; enhanced target cRNN: training with regularization enhancement on the target data; our model: training with transfer learning and regularization enhancement on the source and the target data. **e** Regularization enhancement on manifold learning perspective. Suppose data point without noise (purple hollow circle) concentrate near a low-dimensional manifold. Observed data point (purple solid circle) obtained by corruption process (red arrow) will generally lie farther from the manifold. The model learns with $\tilde{h}$ to project them back onto the manifold by denoising process (blue arrow). **f** The state vector $h_0$ used for generation is the interpolation between $h_{0,1}$ and $h_{0,2}$ of the training data using interpolation algorithm, where the interpolation factor $\alpha$ is a value between 0 and 1. Source data are provided as a Source Data file.

and autoencoder[21]. The architecture of the proposed GDL model is schematically shown in Fig. 2a. Molecules are represented by SMILES strings[20], which are encoded by the "one-hot" representation for inputs and outputs. Combined with the state vectors given by the feature extractor as the conditional input, the cRNN model is trained to generate molecules following the same chemical distribution of given training data in an unsupervised-learning manner. In the training process, the cRNN is trained to reconstruct the input SMILES with regularized state vector as the conditional input; in generating process, the inference cRNN is used to generate molecules triggered by the start token <SOS> with sampling state vector as the conditional input (Fig. 2a). We applied three strategies to enhance the ability to generate molecules against a specific target (RIPK1): transfer learning, regularization enhancement, and sampling enhancement.

Transfer learning. To build a well-performing model from limited known active compounds (target data, such as RIPK1 inhibitors here), we applied transfer learning[13,24] during the training process (Fig. 2b). For general optimization, we pre-trained the generative model using a large-scale dataset containing ~16 million molecules derived from the ZINC12 database[36] (source data). We then fine-tuned the model using the target data (here the target data is comprised of 1030 known RIPK1 inhibitors, Supplementary Table 1). To verify the effect of transfer learning, we evaluated the reconstruction and generation ability using dynamic validation datasets; for these dynamic validation datasets, 100,000 molecules from the source data or 1000 molecules from the target data were randomly selected for reconstruction evaluations, and 100 molecules from either the source data or the target data were randomly selected for generation evaluations. The results showed two remarkable improvements. First, the generalization ability, as assessed by the balanced reconstruction (Fig. 2c) and generation performance (Fig. 2d) on both the source data and the target data, was markedly improved. The models trained only on the target data performed worse on the reconstruction task (Fig. 2c), illustrating the importance of transfer learning. Second, when compared with the models without transfer learning, the convergence time was shortened considerably without affecting the reconstruction accuracy (Supplementary Fig. 1). Therefore, we adopted transfer learning in the subsequent model implementation.

Regularization enhancement. To improve the generation ability of the GDL model, we implemented regularization enhancement[25,26] by randomly adding Gaussian noise to the state vector during model training (Fig. 2e). As a proof of principle of the regularization enhancement, the dynamic validation dataset was evaluated. The results indicated that the GDL model benefitted from regularization enhancement: the enhanced model outperformed the other baseline methods in terms of generation capability while maintaining similar reconstruction performance (Fig. 2c, d).

Sampling enhancement. During the inverse design process of generative models, new molecules are generated by sampling a random state vector in the learned latent space. We adopted sampling enhancement[14,27] to generate new molecules from given state vectors. The performance of three sampling enhancement methods, i.e., single-point sampling, linear-interpolation sampling, and spherical-interpolation sampling, were evaluated on dynamic validation datasets containing 100 randomly selected molecules from the target data. The linear-interpolation sampling (Fig. 2f) outperformed the other two methods (Supplementary Fig. 2). Thus, linear-interpolation sampling was implemented in our framework for molecule generation.

## Generation of a tailor-made virtual compound library for RIPK1 and virtual screening

The GDL model described above was applied to build a tailor-made virtual compound library for RIPK1. By running this model, we obtained a total of 79,323 molecules, in which duplicated molecules in the training sets and molecules bearing structural alerts or reactive groups had already been removed. To visualize the similarity between the source data, the target data, and the generated data in chemical space, uniform manifold approximation and projection (UMAP)[37] plots were generated. As shown in Fig. 3a, the molecules sampled from the generated data (blue) were shifted from the source data (red) toward the target data (purple) after transfer learning, indicating the effectiveness of transfer learning for navigating through chemical space from the source to the target. Moreover, the generated molecules were essentially similar to active compounds (target data) in terms of their physicochemical properties (Fig. 3b and Supplementary Fig. 3). Based on the analysis of relative scaffold diversity (i.e., unique scaffolds/total number of scaffolds)[38], the generated data (26.4%) outperformed the source data (1.2%) and the target data (14.1%) despite the fact that the number of Murcko scaffolds[39] in the source data (193,982) was much

larger than that in the generated data (20,924) (Fig. 3c). Notably, 99.8% and 99.7% of the scaffolds in the generated data were different from the scaffolds in the source data and the target data, respectively, demonstrating the powerful ability of our model to generate additional scaffolds (Fig. 3c). Further, in terms of scaffold diversity[40], the generated data were obviously better than the target data and close to the source data (Fig. 3d and Supplementary Fig. 4a). Regarding fingerprint diversity[40], the generated data were also better than the target data for various types of fingerprints (Supplementary Fig. 4b–g).

We then carried out virtual screening against the generated molecular library to obtain drug-like hit compounds targeted RIPK1. First, in order to ensure the uniqueness of the scaffold, we removed molecules that contain the same generic Murcko scaffolds[39] or the same sub-structures as those in the known RIPK1 inhibitors (target data). Second, drug-like molecule screening was performed according to several important properties associated with drug-like molecules (see the Methods section). Third, pharmacophore-based virtual screening was carried out. To this end, we established a full-feature pharmacophore map[41,42] of RIPK1 inhibitors based on the reported co-crystal structures of RIPK1-ligand, which includes all the important features and interactions between the RIPK1 receptor and ligands. This pharmacophore map consists of 11 features: two hydrogen bond acceptors (A1–A2), three hydrogen bond donors (D1–D3), and six hydrophobic features (H1–H6) (Fig. 3e). Molecules that had at least four features matched with the pharmacophore map were kept. Through the above screening, 23,925 molecules remained, and these filtered molecules still maintain the scaffold and fingerprint diversity as that of the generated data (Fig. 3d and Supplementary Fig. 4). Finally, molecular docking was used to prioritize the filtered molecules. To visually observe the diversity, we generated tree maps (TMAPs)[43] according to RECAP[44]-based structural similarity and molecular properties or docking scores; TMAPs are a technique for unsupervised visualization of high-dimensional data that creates a 2D layout of a minimum spanning tree constructed in the original space. The TMAPs (Fig. 4 and Supplementary Fig. 5) vividly show the diversity and distribution in the chemical space.

From the top-ranked 50 molecules (Supplementary Fig. 6), eight molecules (RI-056, RI-413, RI-470, RI-539, RI-753, RI-962, RI-985, RI-1155) (Supplementary Table 2) with relatively easier synthetic accessibility were chosen to carry out chemical synthesis and bioactivity evaluation; the synthetic accessibility of compounds was judged by our chemical synthesis team. Although the eight molecules were selected according to their synthetic accessibility, they still have a wide distribution in the TMAP (Fig. 4).

## Retrieval of a potent and selective RIPK1 inhibitor

The selected compounds (Fig. 4 and Supplementary Table 2) were chemically synthesized. Given the space limitations, here we only describe the chemical synthesis of RI-962 (Fig. 5a); the chemical syntheses of the other compounds are presented in the Supplementary Information. Commercially available methyl 5-bromo-1-methyl-1*H*-indole-3-carboxylate (**1**) was methylated to give intermediate **2**, which was hydrolyzed and reacted with α-methylbenzylamine to afford intermediate **4**. Intermediate **4** reacted with bis(pinacolato)diboron to give intermediate **5**. The nucleophilic acyl substitution of 7-bromo-[1,2,4]triazolo[1,5-*a*]pyridin-2-amine (**6**) generated intermediate **7**, which then reacted with intermediate **5** by Suzuki–Miyaura reaction to produce compound RI-962.

The obtained compounds were then tested for their kinase inhibitory activity against RIPK1. Four compounds showed activity with half maximal inhibitory concentration ($IC_{50}$) < 10 μM (Supplementary Table 2). Among them, RI-962 was the most potent one with an $IC_{50}$ value of 35.0 nM against RIPK1 (Fig. 5b). The bioactivity of RI-962 was further validated by ADP-Glo assay, which gave an $IC_{50}$ value of 5.9 nM (Fig. 5c).

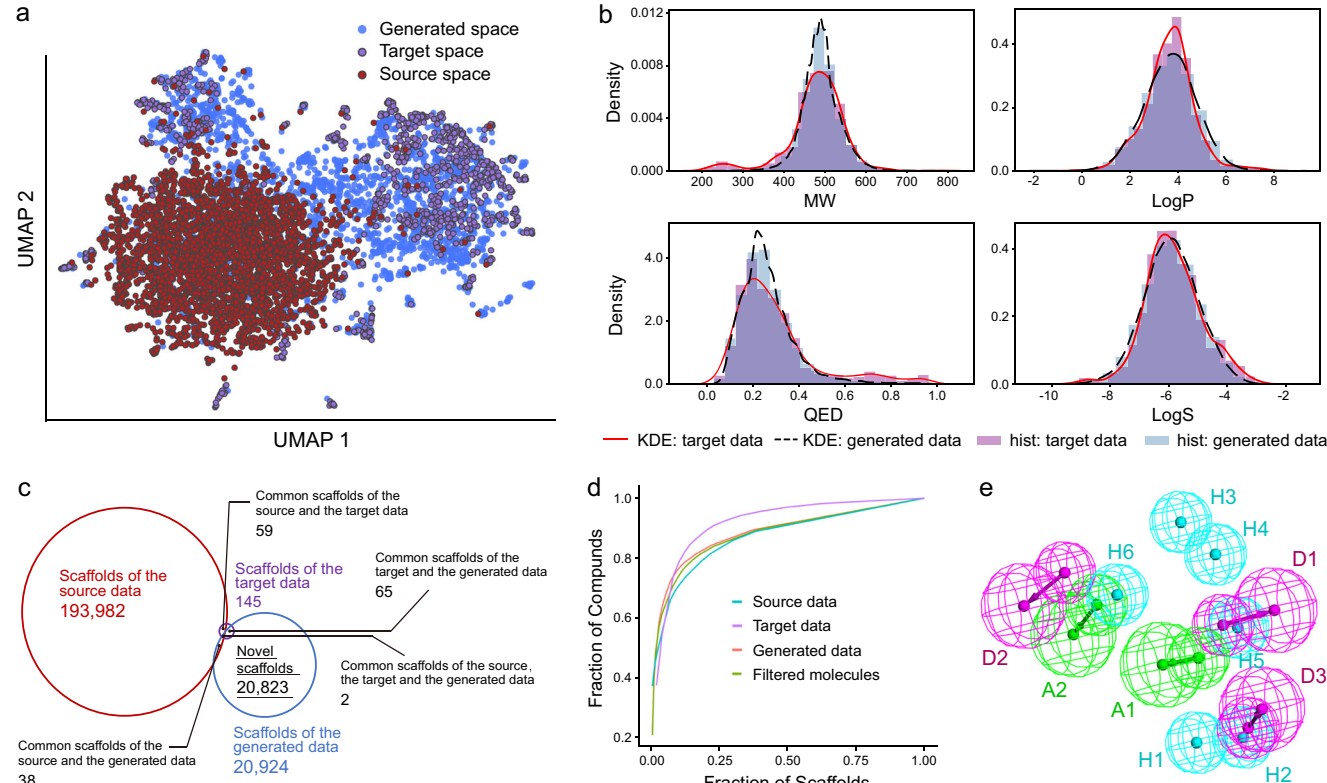

**Fig. 3 | Generation of a virtual compound library against RIPK1 using the cRNN-based generative model. a** Chemical space navigation by transfer learning. UMAP plots of molecule distributions of randomly selected 3000 molecules from the source data, 1000 molecules from the target data and 2000 molecules from the generated data. **b** The distributions of the molecule weight (MW), the water−octanal partition coefficient (LogP), the quantitative estimate of drug-likeness (QED), and the water solubility (LogS) of the target data and the generated data represented by the kernel density estimation (KDE). The KDE demonstrations of the target data and the generated data are displayed by red solid lines and black dotted lines, respectively. Histograms (hist) for the target data and the generated data are represented by purple bar and blue bar, respectively. **c** The Venn diagram of generic Murcko scaffold. The source data, the target data and the generated data are represented by red, purple and blue, respectively. **d** Cumulative scaffold frequency plots showing the distribution of compounds over generic Murcko scaffolds in the source data, the target data, the generated data, and the filtered molecules. **e** The full-feature pharmacophore map of RIPK1. Source data are provided as a Source Data file.

To investigate the kinase selectivity of RIPK1, we performed KINOME*scan* profiling at a concentration of 10 μM against a panel of 408 human kinases (Supplementary Table 3). To kinases that have an inhibitory rate larger than 50%, further $IC_{50}$ values against these kinases were measured. In these assays, RI-962 showed very weak or no activity against all these kinases ($IC_{50}$ > 10 μM) except MLK3, which had an $IC_{50}$ value of 3.75 μM, 107 folds less potent against MLK3 than against RIPK1 (Supplementary Table 4).

### X-ray crystal structure of RIPK1 in complex with RI-962

To understand the potency and selectivity of RI-962, we determined the co-crystal structure of the RIPK1 kinase domain in complex with RI-962 at a solution of 2.64 Å (Supplementary Table 5). As shown in Fig. 6a, RIPK1 adopts its inactive conformation that is characterized by the unique orientation of the conserved Asp-Leu-Gly (DLG) [Asp-Phe-Gly (DFG) in most other kinases] at the base of the activation loop (Fig. 6b). In the inactive conformation (DLG-out), the aspartate side chain of the DLG motif faces into a hydrophobic pocket adjacent to the ATP-binding pocket (called the allosteric site), while its neighboring phenylalanine residue occupies the ATP-binding pocket. In contrast, in the active state (DLG-in), the aspartate faces into the ATP-binding pocket to facilitate catalysis and the phenylalanine side chain occupies the allosteric site. RI-962 occupies both the ATP-binding pocket and the allosteric site simultaneously, indicating a type II kinase inhibitor; kinase inhibitors that occupy the ATP-binding pocket, the allosteric site, or both sites concurrently belong to type I, III or II, respectively.

The triazolo[1,5-*a*] pyridine and indole moieties reside in the ATP-binding pocket and the terminal benzene ring is located in the allosteric site (Fig. 6b). Four hydrogen bonds are formed: the aminotriazole moiety forms two hydrogen bonds with the backbone N and C=O groups of the residue M95; the amide group forms one hydrogen bond with the gatekeeper residue D156, and another hydrogen bond with a water molecule (Fig. 6c).

Compared with the crystal structure of RIPK1 in complex with Cpd8, which is a known type II RIPK1 inhibitor but with poor kinase selectivity, RI-962 induces an evident rotation of the αC-helix in RIPK1 by ~40° (Fig. 6d). Consequently, one of the catalytic triad residues[45], E63, is far away from K45, which breaks the salt bridge interaction between E63 and K45. This rotation also results in a larger empty space in the allosteric site. RI-962 fits snuggly into the re-shaped allosteric site and made tight hydrophobic interactions with resides M67, F162, V134, L129, L70, V75, and I154 (Fig. 6c). Overall, although RI-962 and Cpd8 adopt similar binding poses, RI-962 induces a conformational change around the allosteric site, which leads to a more suitable space in the allosteric site to accommodate RI-962, and better interactions between RI-962 and residues in the allosteric site. This together with the non-conservation of residues in the allosteric site could be used to interpret the high kinase selectivity of RI-962[46].

Nec-1a (Fig. 1) is a highly selective RIPK1 inhibitor, which has often been used as a positive control in necroptosis-related studies[45,46]. The crystal structure of Nec-1a-RIPK1 complex shows that Nec-1a also induces very similar conformational change as RI-962 does and

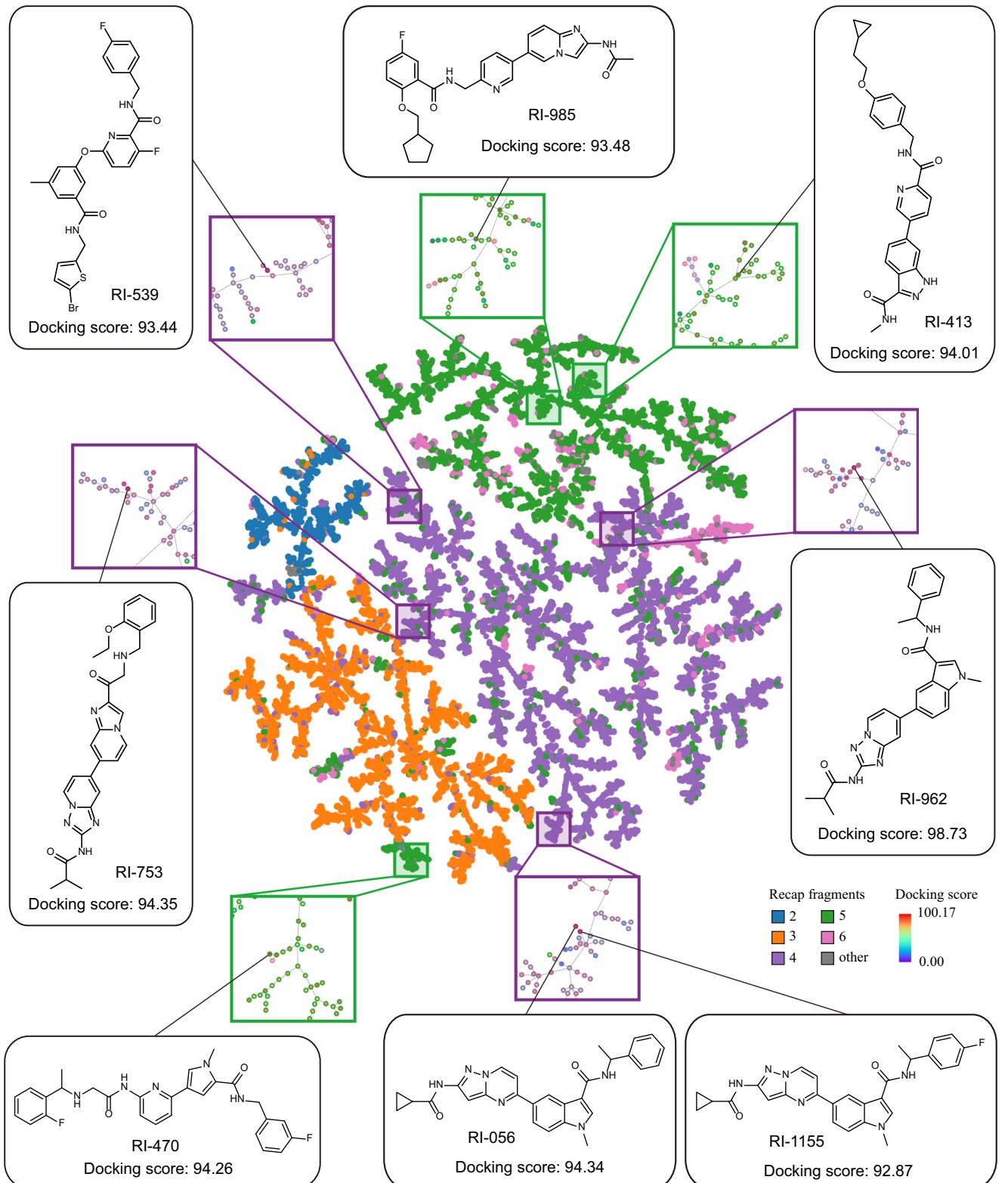

**Fig. 4 | Location of selected molecules for further experimental validation in the TMAP of the filtered molecules.** Center: overview of the TMAP colored by the number of RECAP fragments. Surrounding boxes: the zoom-in versions of their corresponding field with molecules colored by the docking score (red–yellow–green). Molecules with docking scores: selected molecules pointing to their position in the TMAP.

occupies (only) the allosteric site (type III), rendering its kinase selectivity (Fig. 6e). Compared with Nec-1a, RI-962 occupies both the ATP-binding pocket and the allosteric site (Fig. 6e), implying bearing more interactions with RIPK1 and hence presenting higher potency (RI-962, 35 nM vs. Nec-1a, 317 nM[45]).

## Cellular and molecular effects of RI-962

The TSZ (TNFα, Smac mimetic, and Z-VAD-FMK)-induced cell necroptosis models[47] were adopted to examine the cellular effect of RI-962. Four cell lines (HT29, L929, J774A.1, and U937) were used in this assay. As shown in Fig. 7a–d, RI-962 exerted a dose-dependent

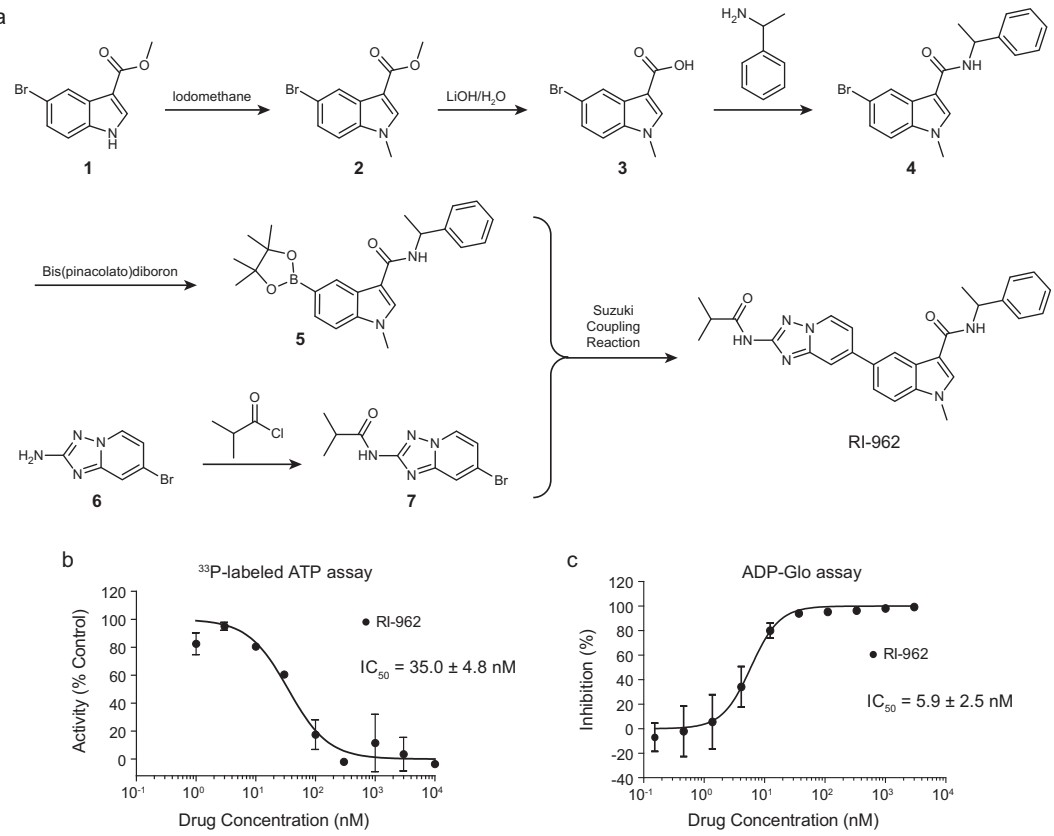

**Fig. 5 | The synthetic route and enzymatic activity of RI-962. a** The chemical structure and synthetic route of RI-962. **b, c** Dose-activity curves of RI-962 in the RIPK1 kinase assay: **b** [33]P-labeled ATP assay at Eurofins ($n = 2$); **c** ADP-Glo assay ($n = 2$). Data are presented as mean ± standard deviation.

protective effect against necroptotic death, with $EC_{50}$ values of 10.0, 4.2, 11.4, and 17.8 nM for HT29, L929, J774A.1, and U937 cells, respectively, which also indicated a cell-independent activity. In addition, the dual staining of HT29 cells with CytoCalcein Violet 450 (for living cells) and 7-AAD (for necrotic cells) visually showed that RI-962 inhibited TSZ-induced necroptosis and improved cell survival in a concentration-dependent manner (Fig. 7e, f). The positive control GSK3145095[48] also displayed activity in these assays, but its potency was relatively weaker compared with that of RI-962. Then, we knocked out RIPK1 in HT29 cells using the CRISPR/Cas9 approach and found that RIPK1 knockout HT29 cells were insensitive to TSZ-induced necroptosis (Fig. 7g, h), implying that RI-962 plays its protective effect against TSZ-induced cell necroptosis by targeting RIPK1.

We next examined the effect of RI-962 on the necroptotic signaling proteins in intact cells. As shown in Fig. 7i, RI-962 markedly inhibited the phosphorylation of RIPK1 and its downstream signaling proteins RIPK3 and MLKL in a dose-dependent manner, whereas it had no effect on the expressions of RIPK1, RIPK3, and MLKL proteins. Again, knockout of RIPK1 had the same effect (Fig. 7j). All these results suggested that RI-962 protects cells from necroptosis by inhibiting the kinase activity of RIPK1.

**Pharmacokinetic characteristics and safety evaluation of RI-962**
To further explore the druggability of RI-962, pharmacokinetic (PK) experiments were conducted in Sprague-Dawley (SD) rats. RI-962 given intravenously (i.v.) (5 mg/kg), intraperitoneally (i.p.) (20 mg/kg) and orally (p.o.) (20 mg/kg) showed the area under the curve ($AUC_{0-t}$) values of 4526.1 h*ng/mL, 6459.7 h*ng/mL, and 1594.9 h*ng/mL, respectively, indicating a proper drug exposure. It displayed a half-life ($T_{1/2}$) of 8.5 h and a bioavailability of 35.7% following i.p. administration. The metabolic stability of RI-962 in rats was good, with a clearance rate (CL) of 18.5 mL/min/kg. (Table 1 and Supplementary Fig. 7).

We further evaluated the maximum tolerated dose of RI-962 in mice, which were well tolerated at doses up to 250 mg/kg, with no observed weight loss and no other side effects (Supplementary Fig. 8).

**In vivo effects of RI-962 in animal models of inflammatory disease**
Necroptosis is associated with a variety of inflammatory disorders, and RIPK1 is considered as a promising intervention target for these diseases[28,30,50]. Thus, we evaluated the in vivo effects of RI-962 in two animal models of inflammatory diseases: TNFα-induced systemic inflammatory response syndrome (SIRS) and dextran sulfate sodium (DSS)-induced inflammatory bowel disease (IBD).

We first examined the in vivo effects of RI-962 on the TNFα-induced SIRS model. SIRS is a life-threatening inflammatory state that results from the complex pathophysiologic response to infection, trauma, burns, pancreatitis, or a variety of other injuries[51]. In this study, a TNFα-induced SIRS mouse model was used to examine the effects of RI-962. As shown in Fig. 8a, a majority of the vehicle-treated mice died within 24 h (survival rate = 10%) after tail vein injection of TNFα. In comparison, the survival rate was increased to 90% in the RI-962-treated group. GSK3145095 also increased the survival rate (50%), but less than RI-962. Treatment with RI-962 or GSK3145095 remarkably reduced the TNFα-induced temperature loss (Fig. 8b) and the concentrations of proinflammatory cytokines (IL-1β and IL-6) in mice (Fig. 8c, d). The hematoxylin and eosin (H&E) staining of heart, liver, spleen, lung, and kidney showed that TNFα injection evidently damaged the liver (as indicated by the inflammatory cell infiltration in the portal area) and kidney (as indicated by a glomerular hemorrhage and swelling with neutrophil infiltration) (Fig. 8e), but had very weak or no obvious impact on heart, spleen and lung (Supplementary Fig. 9). Treatment with RI-962 attenuated damage to the liver and kidney (Fig. 8e). We further explored the mechanism of action by western

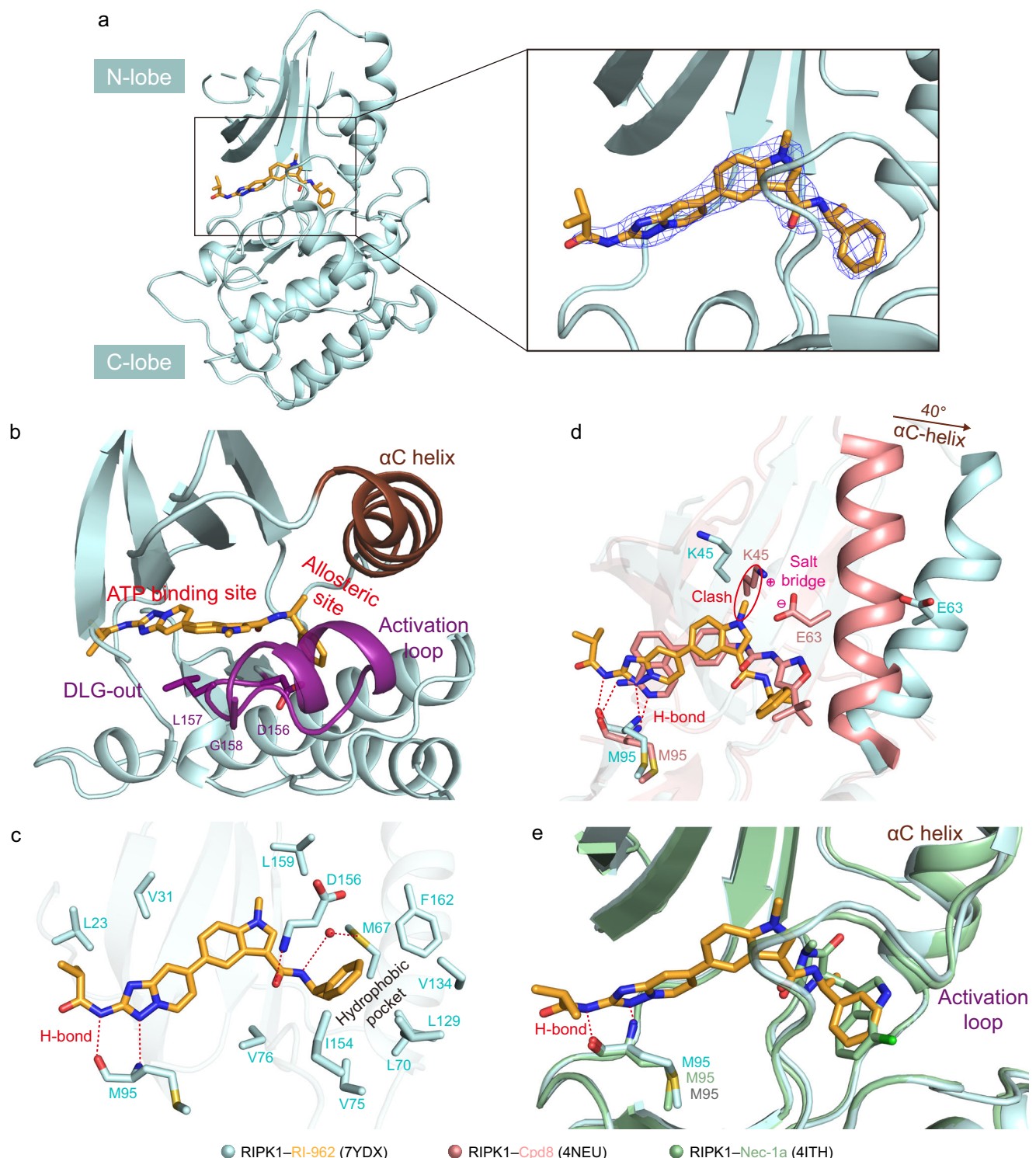

**Fig. 6 | Co-crystal structures of RIPK1 complexed with RI-962 (PDB ID: 7YDX).**
**a** The binding site of RI-962. 2*Fo-Fc* electron density map (contoured at 1.0 σ, blue mesh) is shown for RI-962 (in orange). **b** RI-962-bound RIPK1 adopts an inactive conformation. The αC-helix is colored in chocolate, and the activation loop is colored in deep purple. Position of RI-962 (in orange) in DLG-out structure (sphere, in deep purple) is shown. **c** Interactions between RI-962 and the RIPK1 kinase domain (key residues are shown as pale cyan sticks). **d** Superimposition of the co-

crystal structures of RIPK1 (in salmon)-Cpd8 (in salmon) and RIPK1 (in pale cyan)-RI-962 (in orange). The sidechains of K45 and E63 are shown in sticks.
**e** Superimposition of the co-crystal structure of RIPK1 (in pale green)-Nec-1a (in pale green) and RIPK1 (in pale cyan)-RI-962 (in orange). Hydrogen bonds are represented as red dashed circles. The steric clash is represented as red circle. Salt bridge is represented as positive-negative symbol.

blot. As shown in Fig. 8f, RI-962 treatment substantially reduced the level of phosphorated RIPK1 (pRIPK1) but had no impact on the RIPK1 protein, indicating the inhibition of RIPK1 kinase activity. The activation of downstream proteins, RIPK3 and MLKL, was also markedly

suppressed (Fig. 8f). Taken together, these results indicate that RI-962 ameliorated TNFα-induced SIRS by inhibiting RIPK1 activity.

We then evaluated the in vivo effects of RI-962 on the DSS-induced IBD model. IBD is a chronic, debilitating intestinal disease with

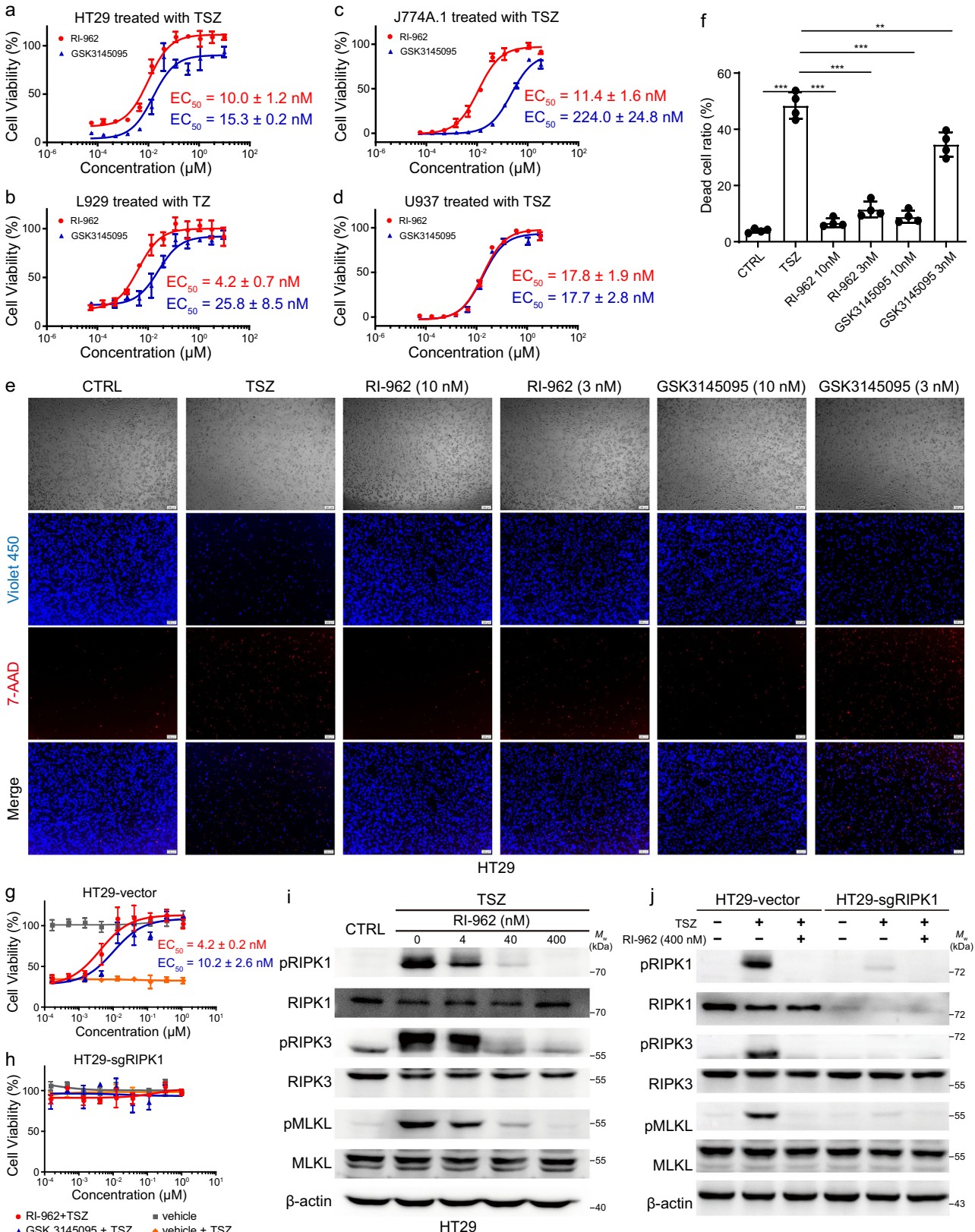

a variety of clinical manifestations. The main forms of IBD are ulcerative colitis and Crohn's disease[52]. Necroptosis is a major type of cell death involved in the regulation of intestinal homeostasis in the intestinal epithelium[53–55]. RIPK1 is thus regarded as a potential target for IBD treatment[56]. In this study, we examined the effect of RI-962 in a DSS-induced IBD mouse model. As shown in Fig. 9a, DSS treatment led

to a rapid loss in mouse body weight from day 5 to day 11, and treatment with RI-962 or GSK3145095 strongly ameliorated this loss of body weight. Further, treatment with RI-962 or GSK3145095 markedly reduced the DSS-induced shortening of colon length (Fig. 9b, c). Histopathological analysis showed that RI-962 substantially decreased tissue damage in the colons of DSS-treated mice (Fig. 9d). In DSS-

**Fig. 7 | RI-962 protected cells from TSZ-induced necroptosis.**
**a**–**d** Dose–response curves of RI-962 ($n = 3$) and GSK3145095 ($n = 2$) in TSZ-induced HT29, L929, J774A.1, and U937 cell necroptosis models. Necroptosis of HT29, J774A.1 and U937 cells was induced by TNFα (10 ng/mL), Smac-mimetic (100 nM), and Z-VAD-FMK (40 μM). Necroptosis of L929 cells was induced by TNFα (10 ng/mL) and Z-VAD-FMK (40 μM). Cell viability was measured by CCK8 staining after 24 h of inducing necroptosis. Data are presented as mean ± standard deviation.
**e** Dual staining with CytoCalcein Violet 450 and 7-AAD in HT29 cells with/without RI-962 and GSK3145095 treatment after TSZ-induced necroptosis for 24 h.
**f** Statistic percentage of dead cells in the dual staining assay ($n = 4$). Data are presented as mean ± standard deviation, **$p$-value < 0.01, ***$p$-value < 0.001 by two-tailed unpaired Student's $t$-test. **g**, **h** Knockout of RIPK1 desensitizes HT29 cells to TSZ-induced necroptosis. Cell viability was measured by CCK8 staining after 24 h of inducing necroptosis ($n = 3$). Data are presented as mean ± standard deviation. **i** RI-962 inhibited the phosphorylation of RIPK1, RIPK3, and MLKL in HT29 cells. β-actin was used as an internal control. Molecular weight ($M$w) markers are shown at right. The experiment was performed three times with similar results. **j** Knockout of RIPK1 inhibited the phosphorylation of RIPK3 and MLKL, and compound RI-962 had the same effect. β-actin was used as an internal control. Molecular weight ($M$w) markers are shown at right. The experiment was performed three times with similar results. Source data are provided as a Source Data file.

induced colitis, numerous S100a9-positive cells (a marker of inflammation) infiltrated into the mucosa and epithelial layer of the damaged colon (Fig. 9e), while no infiltration by S100a9-positive cells was observed in the colons of mice treated with RI-962 (Fig. 9e). More importantly, treatment with RI-962 or GSK3145095 dramatically increased the survival rate of DSS-treated mice (Fig. 9f; 40 mg/kg RI-962 or GSK3145095 survival rate, 100% vs vehicle: 16.7%). In addition, RI-962 treatment during DSS challenge substantially reduced the content of proinflammatory cytokines (TNFα, IL-1β, and IL-6) in cultured colonic tissue supernatants compared with the DSS control mice (Fig. 9g–i). Finally, the western blot assay was used to investigate the effect of RI-962 on the RIPK1 signaling pathway. The results showed that RI-962 reduced the levels of pRIPK1, pRIPK3, and pMLKL proteins in the colon during DSS challenge, but did not impact the expression of RIPK1, RIPK3, and MLKL proteins (Fig. 9j), suggesting that RI-962 suppressed the RIPK1 signaling in the mouse model of DSS-induced colitis.

## Discussion

Developing a new drug is an expensive and time-consuming process that might take over 1 billion dollars and over 10 years. Identification of hit/lead compounds with novel structures is the first and also a critical step. The most common approach to retrieving new hit/lead compounds is to screen existing chemical libraries by using high-throughput screening methods. By this approach, one may not be able to locate additional active compounds with different scaffolds due to the limited chemical space of the existing compound libraries that have already been screened over and over again. To this end, we in this investigation proposed a GDL model to generate a tailor-made compound library with previously unreported scaffolds, which allows us to retrieve hit/lead compounds from the huge unexplored chemical space.

The proposed GDL model is a cRNN-based model[21]. The generative process of cRNN is conditioned by explicitly setting its internal state according to desired properties. Current implementations of cRNN usually employ goal-directed strategies. However, the effectiveness of the goal-directed model strongly depends on the accuracy of the goal function. Ill-defined goal functions can result in invalid molecular structures[18,23]. As an alternative to goal-directed approach, the distribution-learning strategy aims to generate molecules that resemble the given dataset, which could achieve data-driven molecule generation through unsupervised learning[22]. We therefore established a distribution-learning cRNN model, in which three strategies including transfer learning, regularization enhancement, and sampling enhancement were incorporated. Transfer learning shifted the data distribution of the latent space from the large collection of the source data (ZINC12 database) toward the target data (known RIPK1 inhibitors), enabling the generation of drug-like and bioactive molecules. Regularization enhancement by adding random input noise, which is considered equivalent to introducing penalty terms in the objective function[25,26], is beneficial to improve the generalization performance of the GDL model. Sampling enhancement is implemented by interpolating between latent space during model generation, which improves the likelihood of successful generation of target-specific molecules with diverse chemical scaffolds.

Our GDL model has been successfully applied to establish a virtual compound library against RIPK1. The generated library was enriched with much more new scaffold molecules compared with the known RIPK1 inhibitors. Through a standard drug screening process against the established compound library, we retrieved a potent and selective RIPK1 inhibitor with a previously unreported scaffold. On the one hand, this application example verified the effectiveness of our GDL model. Despite that RIPK1 is a kinase, our GDL model could be applied to different kinds of biological targets. The only requirement is that the biological targets must have a sufficient number of known active compounds (target data). The bigger the number of known active compounds is, the better the GDL model is expected to perform. On the other hand, this application example led to the identification of a potent RIPK1 inhibitor (RI-962) with a previously unreported scaffold. Of note is that RI-962 displayed high selectivity against other 407 kinases. It also showed potent activity both in vitro and in vivo. Even so, this compound still has some unfavorable properties that need further optimization in future, for example, low oral bioavailability (Table 1). This situation is understandable because the GDL model is not a panacea and we should not hold an extravagant hope to directly generate a drug candidate by this model. Overall, we discovered a lead compound with a previously unreported scaffold against RIPK1 by using our proposed GDL model, witnessing a successful application of deep neural network in early drug discovery.

## Methods
### Data preparation
Compounds from ZINC12 database[36] were used to construct the source data for transfer learning (downloaded on August 20, 2020).

## Table 1 | Key pharmacokinetic parameters of RI-962 obtained in a preliminary pharmacokinetic assessment experiment[a]

| Parameter | RI-962 | | |
|---|---|---|---|
| | i.v. | p.o. | i.p. |
| Dose (mg/kg) | 5 | 20 | 20 |
| $T_{1/2}$ (h) | 2.1 ± 0.2 | 1.3 ± 0.2 | 8.5 ± 1.6 |
| $T_{max}$ (h) | 0.1 ± 0.0 | 0.8 ± 1.0 | 0.5 ± 0.0 |
| $C_{max}$ (ng/mL) | 12170.4 ± 1198.5 | 674.2 ± 424.7 | 3603.3 ± 693.3 |
| $AUC_{0-t}$ (ng*h/mL) | 4526.1 ± 546.0 | 1594.9 ± 891.8 | 6459.7 ± 1131.6 |
| $AUC_{0-\infty}$ (ng*h/mL) | 4538.1 ± 546.3 | 1604.5 ± 896.1 | 6609.3 ± 1121.4 |
| $V_{ss}$ (L/kg) | 0.4 ± 0.1 | - | - |
| $MRT_{0-\infty}$ (h) | 0.4 ± 0.0 | 1.8 ± 0.2 | 2.8 ± 0.1 |
| CL (mL/min/kg) | 18.5 ± 2.1 | - | - |
| $F$ (%) | - | 8.8 ± 5.0 | 35.7 ± 6.3 |

[a]Data are shown as mean ± standard deviation; $n = 3$ animals.

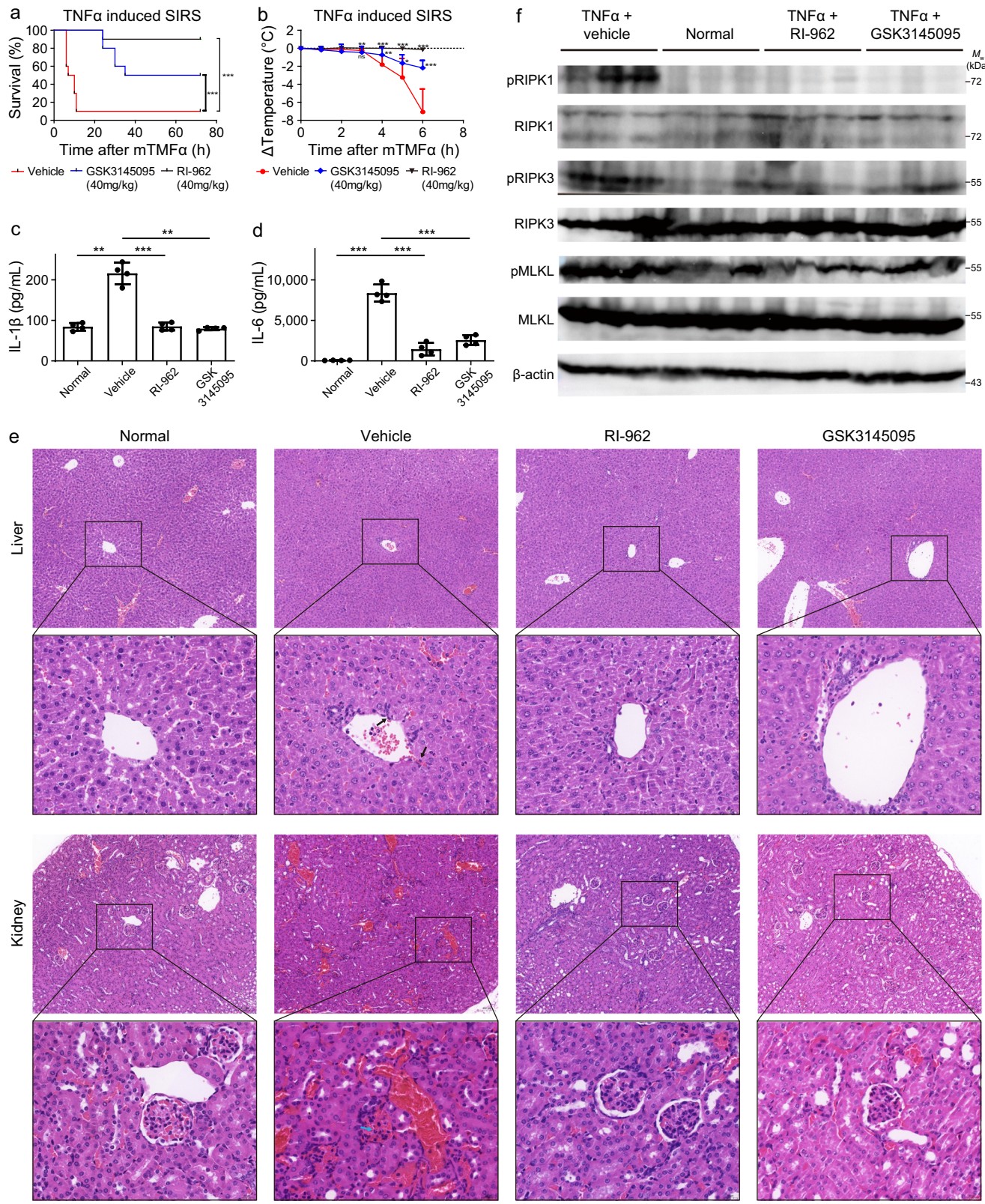

Known RIPK1 inhibitors (bioactive compounds) were retrieved from ChEMBL[57] and patents (<10 μM) to form the target data. All these molecules were encoded as SMILES strings, and then canonicalized and standardized by removing stereochemical information, salts, and duplicates using the RDKit package (v2019.09.2.0, www.rdkit.org). We finally obtained a set of ~16 million molecules as the source data and 1030 bioactive molecules as the target data (Supplementary Table 1).

## Implementation of the GDL model

The generative model reads the input SMILES string[20] of a molecule with "one-hot" representation and a state vector coded by the feature extractor, and then converts them back to the SMILES string following chemical rules. The generative model is a one-layer LSTM (256 dimensions) followed by a dense layer with a SoftMax activation function to generate a probability distribution over all possible

**Fig. 8 | RI-962 ameliorates TNFα-induced SIRS.** C57BL/6 female mice ($n = 14$) were treated with vehicle, RI-962 (40 mg/kg), or GSK3145095 (40 mg/kg) via intraperitoneal injection for 15 min followed by the tail vein injection of mouse TNFα (300 µg/kg). At 6 h after TNFα injection, four mice in each group were killed at random, and the serums, hearts, livers, spleens, lungs, and kidneys were collected for analysis. **a** Survival rates of the mice treated with vehicle ($n = 10$), RI-962 ($n = 10$), and GSK3145095 ($n = 10$) after TNFα injection. \*\*\**p*-value < 0.001 by Gehan-Breslow-Wilcoxon test. **b** Body temperature loss in the mice treated with vehicle ($n = 14$), RI-962 ($n = 14$), and GSK3145095 ($n = 14$) after TNFα injection. Data are presented as mean ± standard deviation, \**p*-value < 0.05, \*\**p*-value < 0.01, \*\*\**p*-value < 0.001 by two-tailed unpaired Student's *t*-test. **c**, **d** The mice ($n = 4$) were killed at 6 h after TNFα administration, and the serum concentrations of IL-1β and IL-6 were measured using ELISA kits. Data are presented as mean ± standard deviation,

\*\**p*-value < 0.01, \*\*\**p*-value < 0.001 by two-tailed unpaired Student's *t*-test. **e** The mice ($n = 4$) were killed at 6 h after TNFα administration, and the liver and kidney tissues were collected for analysis. Representative images of the histological analyses of the liver and kidney tissues by H&E staining (scale bar = 100 µm). Magnified views of the boxed regions for each image are shown below (scale bar = 20 µm). The black arrow represents portal area inflammatory cell infiltration, and the blue arrow represents glomerular neutrophil infiltration. **f** Liver proteins were tested by western blot to detect RIPK1, pRIPK1, RIPK3, pRIPK3, MLKL, and pMLKL with corresponding antibodies. β-actin was used as an internal control. Western blot represents three mice from each group. Molecular weight (*Mw*) markers are shown at right. Data were obtained from two independent experiments. Source data are provided as a Source Data file.

grammar production rules for each time step. The feature extractor is a one-layer bi-directional LSTM (512 dimensions) to convert the input molecule to an initial state vector. In short, a cRNN takes a sequence of input vectors $x_{1:n} = (x_1,...,x_n)$ and an initial state vector $h_0$, and returns a sequence of state vectors $h_{1:n} = (h_1,..., h_n)$ and a sequence of output logit vectors $o_{1:n} = (o_1,..., o_n)$ (Eq. (1)). The model cRNN consists of a recursively defined function $R$ (Eq. (2)), which takes a state vector $h_{i-1}$ and input vector $x_i$ and returns a new state vector $h_i$; another function $O$ maps a state vector $h_i$ to an output logit vector $o_i$ (Eq. (3)):

$$\text{cRNN}(h_0, x_{1:n}) = h_{1:n}, o_{1:n} \tag{1}$$

$$h_i = R(h_{i-1}, x_i) \,, i \geq 1 \tag{2}$$

$$o_i = O(h_i) \,, i \geq 1. \tag{3}$$

During training, we trained the generative model to reconstruct the training data by minimizing training loss $\mathcal{L}$ (Eq. (4)), which was evaluated as the similarity between the original and reconstructed vectors of molecular representations. Training loss $\mathcal{L}$ was computed from the cross-entropy loss function with SoftMax activation (Eq. (4)):

$$\mathcal{L} = \frac{1}{n} \sum_{i=1}^{n} \left( -\sum_{j=1}^{J} y_{i,j} \log p_{i,j} \right) \tag{4}$$

$$p_{i,j} = \frac{e^{o_{i,j}}}{\sum_{k=1}^{K} e^{o_{i,k}}}, \tag{5}$$

where $n$ is the batch size, $J$ is the dimension of each molecular representation, $i$ is the $i$th vector, $j$ is the $j$th dimension of a vector, $k$ is the $k^{\text{th}}$ dimension of a vector, $K$ is the set of all tokens, $y$ is the vector of the original molecular representation (label), and $o$ is the vector of the reconstructed molecular representation. The parameters of the generative model were then updated using AdamOptimizer with learning rate of 0.0001 until convergence. Training loss was monitored and visualized using TensorBoard. Transfer learning[13,24] was implemented by updating the parameters using the target data, based on the parameters of the converged pre-trained model using the source data. Regularization enhancement[25,26] was performed by adding a Gaussian noise vector $\xi$ to the hidden vector $h_0$ (Eq. (6)):

$$h_0^{\text{noise}} = h_0 + \xi \,, \xi_m \in \mathcal{N}(\mu, \sigma^2), \tag{6}$$

where $h_0^{noise}$ is the regularized $h_0$, $\xi$ is the noise vector with the same dimension of $h_0$, $\xi_m$ is the $m^{\text{th}}$ dimension of the noise vector $\xi$, and $\mathcal{N}(\mu,\sigma^2)$ is a Gaussian distribution with mean $\mu$ and variance $\sigma^2$. The mean $\mu$ of the noise distribution was chosen to be zero.

During generation, molecular representations were generated by the start token <SOS> and $h_{\text{new}}$ with sampling enhancement[14,27]. Three

types of sampling enhancement were implemented, namely, linear-interpolation sampling [Linear, Eq. (7)], spherical-interpolation sampling [Slerp, Eq. (8)], and single-point sampling [Sample, Eq. (9)]:

$$h_{0,\text{new}}^{ij,\alpha} = \text{Linear}\left(h_0^i, h_0^j; \alpha\right) = (1 - \alpha)h_0^i + \alpha h_0^j \,, \alpha \in (0,1) \tag{7}$$

$$h_{0,\text{new}}^{ij,\beta} = \text{Slerp}\left(h_0^i, h_0^j; \beta\right) = \frac{\sin[(1 - \beta)\theta]}{\sin\theta} h_0^i + \frac{\sin(\beta\theta)}{\sin\theta} h_0^j \,, \beta \in (0,1) \tag{8}$$

$$h_{0,\text{new}}^i = \text{Sample}\left(h_0^i\right) = h_0^i + \xi_s \,, \xi_s \in \mathcal{N}\left(\mu_s, \sigma_s^2\right), \tag{9}$$

where $\alpha$ is the linear-interpolation factor, $\beta$ is the spherical-interpolation factor, $\theta$ is the central angle of $h_0^i$ and $h_0^j$, and $\xi_s$ is a random vector that has the same dimension as $h_0^i$ and belongs to the Gaussian distribution with mean $\mu_s$ and variance $\sigma_s^2$.

All the parameters used in the GDL model are presented in Supplementary Table 6. All software programs were implemented in Python (v3.6.9) with the TensorFlow GPU backend (www.tensorflow.org, v1.10.0). Additional details are provided in Supplementary Information, including conversion between SMILES and word embedding matric (Supplementary Note 1), regularization enhancement (Supplementary Note 2), and sampling enhancement (Supplementary Note 3).

### Evaluation of the GDL model

The performance of the GDL model was evaluated on subsets randomly selected from the source data or the target data. To evaluate the reconstruction capability of the GDL model, we used 100,000 molecules from the source data and 1000 molecules from the target data as subsets, and the criterion was the reconstructed rate ($R\%$, Eq. (10)):

$$R\% = \frac{M_{\text{recon}}}{N_{\text{recon}}} \times 100\%, \tag{10}$$

where $N_{\text{recon}}$ is the number of subset molecules used for evaluation of reconstruction capability, and $M_{\text{recon}}$ is the number of molecules that are reconstructed correctly by the GDL model. We evaluated the performances of models trained using six training methods: (1) training on the source data; (2) training on the target data; (3) training with transfer learning on the source and the target data; (4) training with regularization enhancement on the source data; (5) training with regularization enhancement on the target data; and (6) training with transfer learning and regularization enhancement on the source and the target data. To further evaluate the generation capability, we used 100 molecules from either the source data or the target data, respectively. The models were trained using four training methods with qualified reconstructed capability: (1) training on the source data; (2) training with transfer learning on the source and the target data; (3) training with regularization enhancement on the source data; and (4) training with transfer learning and regularization enhancement on the

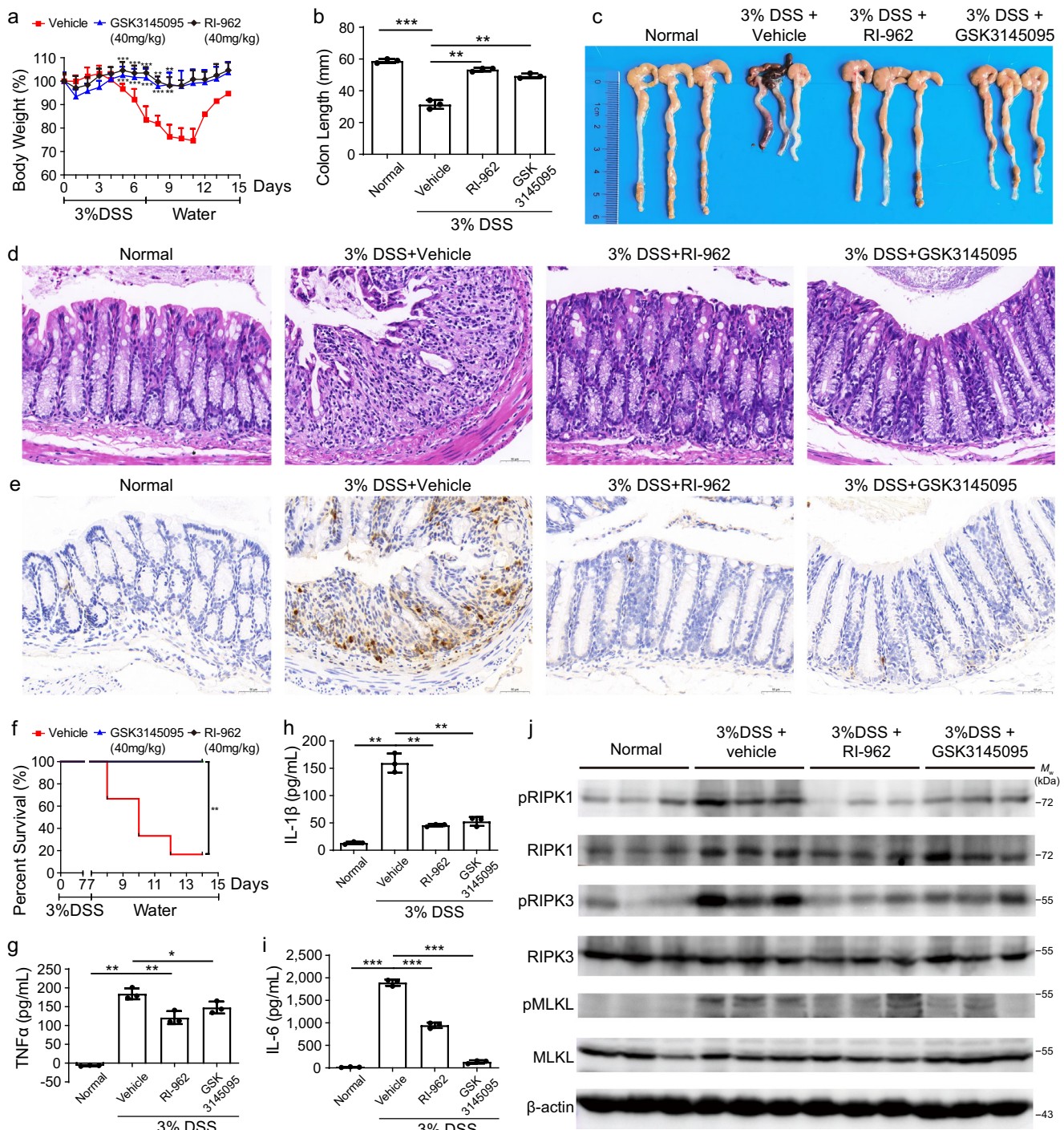

**Fig. 9 | RI-962 reduces inflammation in acute DSS-induced colitis.** DSS (3%) was administered to C57BL/6 female mice via drinking water for 7 d. After 7 d, the DSS water was replaced with fresh water. Vehicle, RI-962 (40 mg/kg), or GSK3145095 (40 mg/kg) was injected intraperitoneally once a day for 10 d. **a** Body weight changes in the mice treated with vehicle ($n = 9$), RI-962 ($n = 10$), and GSK3145095 ($n = 10$) after the DSS induction of colitis. Data are presented as mean ± standard deviation, **$p$-value < 0.01, ***$p$-value < 0.001 by two-tailed unpaired Student's $t$-test. **b, c** The mice ($n = 3$) were sacrificed after the DSS induction of colitis for 7 d, and the lengths of the colons from each group of mice were measured. Data are presented as mean ± standard deviation, ***$p$-value < 0.001 by two-tailed unpaired Student's $t$-test. **d, e** The mice ($n = 3$) in each group were sacrificed on day 7, and the colon tissues were collected for analysis. **d** Representative H&E staining of the colon tissues from the mice on day 7 (scale bar = 50 μm). **e** Representative immunohistochemical staining of colon tissues harvested on day 7 and stained for S100a9 with the corresponding antibodies (scale bar = 50 μm). **f** Survival rates of mice treated with vehicle ($n = 6$), RI-962 ($n = 7$), and GSK3145095 ($n = 7$) after the DSS induction of colitis. **$p$-value < 0.01 by Gehan-Breslow-Wilcoxon test. **g–i** On day 7, three mice ($n = 3$) in each group were killed, and the TNFα, IL-6, and IL-1β concentrations in the supernatant of the cultured colon tissues were measured by ELISA. Data are presented as mean ± standard deviation, *$p$-value < 0.05, **$p$-value < 0.01, ***$p$-value < 0.001 by two-tailed unpaired Student's $t$-test. **j** Colonic proteins on day 7 were tested by western blot to detect RIPK1, pRIPK1, RIPK3, pRIPK3, MLKL, and pMLKL with corresponding antibodies. β-actin was used as an internal control. Western blot represents three mice from each group. Molecular weight ($M_w$) markers are shown at right. Data were obtained from two independent experiments. Source data are provided as a Source Data file.

source and the target data. The generation capabilities of these trained models were then evaluated using the generative rate ($G\%$, Eq. (11)) as a criterion:

$$G\% = \frac{M_{\text{gen}}}{N_{\text{gen}}} \times 100\%, \qquad (11)$$

where $N_{\text{gen}}$ is the number of subset molecules used for the evaluation of generation capability, and $M_{\text{gen}}$ is the number of molecules generated by the GDL model.

## Calculations of molecular properties

To compare the similarity between different molecules in terms of physicochemical properties, some important physiochemical parameters associated with drug-like properties were calculated, including molecular weight (MW), the water–octanal partition coefficient (LogP)[58], the qualitative estimate of drug-likeness (QED)[59], Bertz $C_{\text{T}}$[60], the topological polar surface area (TPSA)[61], water solubility (LogS)[62], the number of rotatable bonds (rot), the number of H-bond donors (HBD), and the number of H-bond acceptors (HBA). To visualize the comparison results, histograms and kernel density estimation (KDE) maps were drawn using Seaborn (https://seaborn.pydata.org/, v0.11.1). For the drug-like compound screening, we used the following criteria: $200 \leq MW \leq 700$, $-2 \leq LogP \leq 6$, and $0.15 \leq QED$. To avoid molecules that are very difficult to synthesize, we calculated the synthetic accessibility (SA) score[63] and filtered out molecules with SA score > 5; the SA scores indicate the complexity for synthesis, which ranges from lower values (easy to synthesis) to high values (difficult to synthesis). All calculations were carried out by using RDKit (https://rdkit.org/, v2019.09.2.0).

## Uniform manifold approximation and projection (UMAP)

To visualize the similarity relations between the source data, the target data, and the generated data, we constructed UMAP plots[37] (umap-learn 0.4.6), which are two-dimensional representations of high-dimensional data distributions, from 3000, 1000, and 2000 randomly selected molecules from the source data, the target data, and the generated data, respectively.

## Scaffold and fingerprint diversity

Scaffold and fingerprint diversity were analyzed and visualized using the Platform for Unified Molecular Analysis (PUMA)[40] (https://www.difacquim.com/d-tools/) with 1000 molecules randomly selected from the source data, the target data, the generated data, and the filtered molecules, respectively.

## Full-feature pharmacophore map

The Discovery Studio (version 3.1) program package was used to generate a full-feature pharmacophore map for RIPK1 inhibitors. 13 crystallographic structures of RIPK1-inhibitor complexes were collected from the protein data bank (PDB)[64] (Supplementary Note 4). Taking 4NEU as the reference structure, the MODELER was used for structural alignment with default parameters settings. We then performed the Receptor-Ligand Pharmacophore Generation protocol for the automatic construction of three-dimensional pharmacophores based on the previously aligned structures. All the identified pharmacophore features including hydrogen bond donor (HDB/D), hydrogen bond acceptor (HAD/A), hydrophobic (HYD/H), positive ionizable (PI), negative ionizable (NI), ring aromatic (RA), and excluded volume features were clustered according to their interaction pattern with the receptor. Finally, 11 clustered features including two hydrogen bond acceptors (A1–A2), three hydrogen bond donors (D1–D3), and six hydrophobic features (H1–H6) were selected to form the full-feature pharmacophore map[41,42]. After generating multiple molecular conformations, the screening procedure was carried out, resulting in a set

of molecules with at least four matched pharmacophore features ranked based on their fit values. More details are provided in Supplementary Note 4.

## Molecular docking

The GOLD program was adopted for molecular docking with Gold-Score being used as the scoring function[65,66]. To achieve a better screening, flexible docking was performed. The receptor structure was taken from the protein data bank (PDB)[64] (PDB entry: 4ITH). In order to accelerate flexible docking, we set limited residues to be flexible. By comparison between X-ray crystal structures of RIPK1 in complex with different ligands, we found that, among all the residues forming the active pocket (including the ATP-binding pocket and the allosteric site), nine residues often display a large displacement, including V31, I43, K45, M67, L70, M92, L157, L159, and F162. Therefore, the sidechains of the nine residues were defined as flexible sidechains in the program setting. The binding site was defined as the area within 10 Å around the 4ITH ligand, and other parameters were set to default values. The entire process of molecular docking was implemented in Discovery Studio 3.1.

## Tree maps (TMAPs)

For the unsupervised visualization of high-dimensional data, a TMAP[43] (tmap 1.0.4; faerun 0.3.20) creates a two-dimensional layout of a minimum spanning tree constructed in the original space. In this study, TMAPs were used to visualize RECAP[44]-based (rdkit 2019.09.2.0) structural similarity among the filtered molecules. Each TMAP shows the molecules as dots with up to three concentric circles: the first circle depicts the molecule properties (including MW, LogP, SA score, and QED) or docking scores (colored from red to yellow to green, moving from the maximum value to the minimum value); the second circle depicts the RECAP fragment number of a molecule (colored by the number of RECAP fragments).

## Chemical synthesis

The primary synthetic data are available in the Supplementary Methods.

## Cell lines and cell culture conditions

The cell lines used in this investigation were purchased from the American Type Culture Collection (ATCC). HT29, L929, HEK 293T and J774A.1 cells were cultured in DMEM (Gibco) supplemented with 10% fetal bovine serum, 100 U/mL penicillin, and 100 U/mL streptomycin. U937 cells were cultured in RPMI-1640 (Gibco) culture medium supplemented with 10% fetal bovine serum, 100 U/mL penicillin, and 100 U/mL streptomycin. Sf9 cells were cultured in SIM SF (Sino Biological Inc.) supplemented with 50 U/mL penicillin, and 50 U/mL streptomycin. HT29, L929, HEK 293T, U937 and J774A.1 cells incubations were performed at 37 °C under 5% CO2. Sf9 cells incubations were performed at 27 °C. All cells were negative for mycoplasma, and these cell lines are not among those commonly misidentified by International Cell Line Authentication Committee (ICLAC).

## Cell necroptosis protection assay

Cell necroptosis protection assays were performed in 96-well cell culture plates. Cells were plated in each well and cultured at 37 °C overnight. HT29, U937, and J774A.1 cells were treated with 10 ng/mL TNFα, 100 nM Smac mimetic, and 40 μM z-VAD-FMK for 24 h. L929 cells were treated with 10 ng/mL TNFα and 40 μM z-VAD-FMK for 24 h. The cell survival rate was determined using a CCK8 cell viability assay kit and CLARIOstar (v5.61). The concentration–response curve was fitted using Graph-Pad Prism 8.0 (GraphPad Software) to calculate the 50% effective concentration ($EC_{50}$). All experiments were performed at least two times, and each $EC_{50}$ value was expressed as mean ± standard deviation (SD).

## Dual staining with CytoCalcein Violet 450 and 7-AAD

The assay of dual staining with CytoCalcein Violet 450 and 7-AAD was performed in a 24-well cell culture plate. Cells were plated in each well and cultured at 37 °C overnight. HT29 cells were treated with 10 ng/mL TNFα, 100 nM Smac mimetic, and 40 μM z-VAD-FMK for 24 h. Then use CytoCalcein Violet 450 and 7-AAD to double stain the HT29 cells and observe under the microscope. All images were acquired with an Eclipse Ci-L microscope (Nikon, Japan). The dead cells were measured with Image J software, and the data was analyzed with GraphPad software. The experiment was repeated three times.

## CRISPR/Cas9-mediated RIPK1 knockout in HT29 cells

The lentiCRISPRv2 vector targeting RIPK1 (sgRNA, 5′-CTCGGGCGCCATGTAGTAGA-3′) was constructed by the Azenta company. HEK 293 T cells were transfected with lentiCRISPRv2 targeting RIPK1 and empty vector using Hieff Trans™ Liposomal Transfection Reagent (Yeasen), respectively. The viruses were collected at 24 h and 48 h, respectively, filtered with a 0.45 mm filter head, and then added to the virus concentrate and treated at 4 °C overnight. The concentrated viruses were added to HT29 cells along with 8 μg/mL of polybrene (Yeasen) to enhance transfection efficiency. The infection assay was repeated in the next day under the same conditions. Finally, HT29 cells were screened with 3 μg/mL puromycin. Western blot analysis was used to confirm the RIPK1 deletion.

## Western blot analysis

Cell pellets were collected and resuspended in RIPA lysis buffer (Beyotime), to which phenylmethylsulfonyl fluoride, a proteasome inhibitor, and a phosphatase inhibitor cocktail (Sigma) had been added. Whole-cell protein lysates were incubated on ice for 15 min and centrifuged at $13,800 \times g$ and 4 °C for 15 min. The supernatants were collected and subjected to western blot analysis.

The liver in the SIRS model and the colon in the IBD model were harvested, homogenated and sonicated in RIPA lysis buffer. The supernatants were collected after centrifuged at $13,800 \times g$ and 4 °C for 15 min.

The cell proteins or tissue proteins were separated in a polyacrylamide gel and transferred to a methanol-activated polyvinylidene fluoride membrane. The membrane was blocked for 2 h in Tris-buffered saline plus Tween-20 containing 5% milk and then immunoblotted sequentially with primary and secondary antibodies. Detection was performed with an ECL chemiluminescence kit (Abbkine). The antibodies used were human RIPK1 antibody (R&D, 334640, 1:1000), mouse RIPK1 antibody (Affinity, DF2642, 1:1000), human phospho-RIP (Ser166) rabbit mAb (Cell Signaling Technologies, 65746, 1:1000), mouse phospho-RIP (Ser321) rabbit mAb (Cell Signaling Technologies, 38662, 1:1000), human RIPK3 (B-2) antibody (Santa Cruz, sc-374639, 1:250), mouse RIPK3 antibody (Abcam, ab62344, 1:1000), human anti-RIP3 (phospho S227) antibody (Abcam, ab209384, 1:2000), mouse anti-RIP3 (phospho T231 + S232) antibody (Abcam, ab205421, 1:500), anti-MLKL (58–70) antibody (Sigma, M6697, 1:250), human anti-MLKL (phospho S358) antibody (Abcam, ab187091, 1:1000), mouse anti-MLKL (phospho S345) antibody (Abcam, ab196436, 1:1000), and β-actin (Proteintech, 66009-1-Ig, 1:1000).

## Protein preparation and crystallization

The RIPK1 protein expression and purification were carried out following the similar protocols as those in literature[45]. The human RIPK1 kinase domain containing residues 1–294 with four cysteine-to-alanine mutations (C34A, C127A, C233A, and C240A) was cloned into the vector pFastbacHAT (completed by the Azenta company). The recombinant virus containing RIPK1 was generated using the Bac-to-Bac baculovirus expression system and infected Sf9 cells. After infection by baculoviruses for 48 h, the cells were harvested in a buffer containing 25 mM Tris (pH 7.6), 1 M NaCl, 0.5 mM TCEP, and 20 mM imidazole. The RIPK1 kinase domain was purified to homogeneity using a nickel resin column. The protein was eluted in buffer containing 250 mM imidazole. The N-terminal tag was cleaved by TEV protease, and the protein was further purified using a Superdex 200 gel filtration column (GE Healthcare) and finally using a MonoQ column (GE Healthcare). The purified RIPK1 was concentrated to 10.693 mg/mL in a buffer containing 25 mM Tris-HCl pH 7.9, 150 mM NaCl, and 0.5 mM TCEP.

Crystals of the RIPK1 in complex with RI-962 (final concentration of 1 mM added to the protein) were obtained by co-crystallization via hanging drop vapor diffusion. Crystals were obtained from solution (0.25 M $NH_4I$, 23% polyethylene glycol 3350, and 0.03 M glycyl-glycyl-glycine) and grew to full size in ~1 week. The crystals were harvested after cryo-protection in 10% ethylene glycol and flash-frozen in liquid nitrogen for data collection.

## Data collection and refinement of RIPK1

All diffraction datasets were collected on beamline BL19U1 of the Shanghai Synchrotron Radiation Facility and processed using HKL2000[67]. Further data processing was carried out using programs from the CCP4 suite[68]. Structures were determined by molecular replacement using a previously published structure (PDB ID: 4ITJ)[45] as the starting model. Manual model rebuilding and refinement were iteratively performed with *Coot*[69] and *Phenix*[70], respectively. The crystal of RI-962-bound RIPK1 is in the space group, $P2_12_12_1$. Each asymmetric unit contains two molecules of RIPK1. The statistics and refinement values of the crystal structure are shown in Supplementary Table 5.

## In vitro kinase activity assays

In vitro kinase activity assays were conducted through the Kinase Profiling Services provided by Eurofins (Eurofins, France). The protocol for the RIPK1 assay is briefly described as follows (Protocols for other kinases are very similar and can be found in http://www.eurofins.com/pharmadiscovery). RIPK1 kinase was incubated with the test compound in assay buffer containing 8 mM MOPS (pH 7.0), 0.2 mM EDTA, 250 μM KKKSPGEYVNIEFG, 10 mM magnesium acetate, and 10 μM [γ-33P]-ATP for 15 min at room temperature. The reaction was initiated by the addition of the Mg/ATP mixture. After incubation for 40 min at room temperature, and the reaction was stopped by the addition of 3% phosphoric acid. A 10 μL portion of the reaction mixture was then spotted onto a P30 filter mat and washed four times for 4 min in 0.425% phosphoric acid and once in methanol prior to drying and scintillation counting.

## Source of animals

C57BL/6 mice were purchased from GemPharmatech Co., Ltd. All mice were bred under standard conditions and used at the age of 6–8 weeks when the body weight was ~20 g. All procedures related to animal handling, care and treatment in in vivo efficacy studies were performed according to the guidelines approved by the Institutional Animal Care and Use Committee (IACUC) of West China Hospital, Sichuan University (20211062A). All procedures related to animal handling, care and treatment in pharmacokinetic (PK) studies were performed according to the guidelines approved by the Institutional Animal Care and Use Committee (IACUC) of Shanghai Medicilon Inc.

## The TNFα-induced SIRS experiment

C57BL/6 female mice were first fasted for 12 h (given water) and then the C57BL/6 female mice were pretreated with vehicle, RI-962 (40 mg/kg), or GSK3145095 (40 mg/kg; GSK3145095 was purchased from NewCompoundMarket Pharmatech Co. Ltd.) via intraperitoneal injection for around 15 min and then challenged with mouse TNFα (300 μg/kg) via tail intravenous injection. The body temperatures of

the mice were continuously monitored until 6 h after TNFα administration. At 6 h after TNFα injection, four mice in each group were killed at random, and the serums, heart, liver, spleen, lung, and kidney tissues were collected for analysis. Mice mortality was continuously monitored until 72 h after TNFα administration.

## The DSS-induced IBD experiment
DSS (3% w/v) was administered in drinking water ad libitum for 7 d (from day 0 to day 7). DSS solution was replaced three times on day 2, day 4, and day 6. C57BL/6 female mice were injected intraperitoneally with vehicle, RI-962 (40 mg/kg), or GSK3145095 (40 mg/kg) for 10 d (from day 0 to day 9). Three mice in each group were killed at random on day 7, and distal colon tissues were collected for analysis. The mice weight and survival rate were recorded daily.

## Assessment of pharmacokinetic (PK) properties
The PK properties of compounds were examined in male Sprague-Dawley rats ($n = 3$ per group, weight: 180–220 g). Compounds were dissolved in saline with 5% (v/v) DMSO plus 40% (v/v) PEG400. The animals were administered with a single dose of 5 mg/kg (intravenous injection (i.v.)), 20 mg/kg (intraperitoneal injection (i.p.) or oral gavage (p.o.)). Blood samples were collected at 0.083, 0.25, 0.5, 1, 2, 4, 6, 8, 10 and 24 h, and centrifuged to isolate plasma. Subsequently, the plasma compound concentrations were determined by LC-MS/MS-13 (TQ5500, SCIEX), and the PK parameters were calculated using Phoenix WinNonlin 7.0.

## Enzyme-linked immunosorbent assay (ELISA)
In the TNFα-induced SIRS model, at 6 h after TNFα injection, four mice in each group were killed at random, and the serums were collected, the serum concentrations of IL-1β and IL-6 were measured using ELISA kits (Neobioscience Technology) according to manufacturer's instructions. On day 7 of experimental DSS-induced colitis, the distal colon tissues were harvested, washed with PBS, sliced into small pieces with sizes of ~1 mm³, and cultured with serum-free RPMI-1640 medium (1 mL/100 mg colon tissue) for 12 h. The supernatant was collected by sequential centrifugation at $500 \times g$ for 10 min and $3000 \times g$ for 10 min. The concentrations of cytokines TNFα, IL-6, and IL-1β were measured using ELISA kits (Neobioscience Technology).

## Histological analysis and immunohistochemistry staining
The heart, liver, spleen, lung, kidney and colon tissues were fixed directly in 4% paraformaldehyde (24 h), embedded in paraffin, and stained with H&E following standard procedures. All images were acquired using a Pannoramic MIDI scanner.

The colon tissues were fixed in 4% paraformaldehyde for 24 h. The tissues were sliced to a thickness of 5 μM, deparaffinized with xylene, and rehydrated with graded ethanol. The tissue sections were then placed in a repair box filled with citric acid (pH 6.0) antigen retrieval buffer for antigen retrieval in a microwave oven followed by the quenching of endogenous peroxidase activity in 3% hydrogen peroxide. The sections were incubated overnight at 4 °C with primary antibody (S100a9, 73425, CST), which was prepared in PBS (pH 7.4) according to the manufacturer's instructions. The sections were then washed three times with PBS, incubated for 1 h with the appropriate secondary antibodies, and staining with freshly prepared DAB color developing solution. Subsequently, the sections were counterstained with hematoxylin and mounted in non-aqueous mounting medium. All images were acquired using a Pannoramic MIDI scanner.

## Statistical analysis
Data on figures represent mean ± standard deviation (SD). Unless otherwise noted, the differences between two groups were analyzed by unpaired Student's $t$-test, and differences with $p$-value < 0.05 were considered significant.

## Reporting summary
Further information on research design is available in the Nature Portfolio Reporting Summary linked to this article.

## Data availability
The SDF file of the generated data has been deposited in the Zenodo repository under https://doi.org/10.5281/zenodo.6451205. The crystal structure of the RIPK1–RI-962 complex has been deposited in the Protein Data Bank (PDB) under accession code 7YDX. The crystal structures of RIPK1 used in this study are available in the Protein Data Bank (PDB) under accession codes 4ITJ, 4ITI, 4ITH, 4NEU, 5HX6, 5TX5, 6C4D, 6HHO, 6NW2, 6R5F, 6OCQ, 6NYH, and 6RLN. All other data that support the conclusions are available from the corresponding authors on reasonable request. Source data are provided with this paper.

## Code availability
Computer codes of our GDL model are provided as Supplementary Software and have been deposited in the Zenodo repository under https://doi.org/10.5281/zenodo.7074218.

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

## Acknowledgements

This work was supported by the National Natural Science Foundation of China (T2221004, S.Y.; 81930125, S.Y.; 81773633, S.Y.; 61876034, J.Z.; 22207080, Y.L.), National Postdoctoral Program for Innovative Talents of China (BX2021204, Y.L.), China Postdoctoral Science Foundation (2021M702374, Y.L.),1.3.5 project for disciplines of excellence, West China Hospital, Sichuan University (ZYXY21001, S.Y.; ZYGD18001, S.Y.) and Sichuan University postdoctoral interdisciplinary Innovation Fund. We also thank the staff of the Shanghai Synchrotron Radiation Facility (SSRF) beamlines (Shanghai, China) for great support.

## Author contributions

S.Y. conceived and supervised the research and designed the experiments; S.Y., Y.W., and J.Z. established the GDL model and performed virtual screening. Y.L., R.Y., H.X., and F.W. performed chemical syntheses, separation, purification, and structural characterizations. L.Z., X.L., and C.W. performed gene expression and protein purification, crystallization, diffraction data collection, and crystal structure determination. L.Z., X.L., W.Y., and C.T. performed cellular assays and in vivo studies. S.Y., Y.L., Y.W., L.Z., L.L., and X.Y. analyzed the data. S.Y., Y.L, Y.W., and L.Z. wrote the manuscript.

## Competing interests

The authors declare the following competing interests: Sichuan University has applied for Chinese patents of this work, covering GDL model (application number: 202210883279.7; S.Y., Y.W., J.Z., and X.Y.) and compounds including RI-962 and related compounds (application number: 202110426935.6; S.Y.). The remaining authors declare no other competing interests.
