## [Peer Review File · Nature Communications]

Generative Deep Learning Enables the Discovery of a Potent and Selective RIPK1 InhibitorREVIEWER COMMENTS

Reviewer #1 (Remarks to the Author):

The manuscript entitled "Generative Deep Learning Enables the Discovery of a Potent and Selective RIPK1 Inhibitor" studied a distribution-learning conditional recurrent neural network model to generate tailor-made virtual screening libraries for given biological targets is proposed. The model was then applied to RIPK1. Their contribution is good and the paper.

1- The review of related work is not sufficiently thorough and not sufficiently specific. The authors do not discuss important aspects of the cited works, such as their relative advantages/disadvantages, the results that were reported, or how the different techniques compared to one another and their proposal. Also, although the referenced works are related in terms of the application discussed, this section should be focused on works that employ techniques that are much more similar to the one presented. It should not be difficult to focus on those works that are much more closely related to the methodology presented in the paper and discuss these in-depth.

2- The structure of the paper should be added to the end of the introduction.

3- The proposed approach is described quite superficially, with a noticeable absence of references in which the reader could find the details of the different techniques involved. Publication in a top journal such as *Nature Communications* requires more than just intuitions regarding what a proposed method should achieve: there should be in-depth explanations and formal discussions about how and why the method works, perhaps including experimental evaluation or proofs for each separate module. A comparative discussion of the -detailed- proposal against other similar methods can also be expected. A discussion of the challenges encountered and examples in which the proposed method may fail would also improve the manuscript; in its present form, it would appear that the formulation and application of the proposed method are straightforward, that there are no variables and no possible conditions that may have been unaccounted for by the authors that could alter its performance.

4- The computational details are not mentioned. Which software, program languages, libraries, etc. were used to build this approach? Is it possible to publish code to use this architecture and reproduce the results?

5- The authors describe the used methods in detail. However, this is more literature information. It would enhance the quality of the paper, if more information about the implementation and algorithms of the novel workflow would be provided, too.

6- Lack of coherence, clarity, and poor consistency (e.g., reference is not in the same format).

7- The motivation part can be elaborated in the abstract, in the introduction as well as in the conclusion, if possible, which will enable the authors to improve the academic merit of their work.

8- A brief description of hyperparameter optimization (if done), architecture selection (n. of layers), and training time should be provided.

9- Fig 3 quality needs to be improved.

10- In the Method part, only the notations are given. Please state the problem in detail.

11- The proposal is correct and the results are well-founded. There are some fragments of the Discussion that seem more like a summary. I suggest that the Discussion be rewritten to better reflect the quality of the work.

12- The statement and description of future works are needed.

13- I suggest to the authors that the manuscript be reviewed by a native speaker to ensure the quality of the writing and the spelling

Reviewer #2 (Remarks to the Author):

In the current manuscript, Li et al describe an innovative approach of using a distribution-learning conditional recurrent neural network model to generate tailor-made virtual screening libraries for given

biological targets and apply this model to RIPK1 in a prove-of-concept approach. Using this approach the authors describe the identification of four compounds that showed activities against RIPK1 one of which was then selected based on superior specificity for RIPK1. The authors show data to demonstrate that their candidate inhibits necroptosis and ameliorates disease in inflammatory mouse models. Overall, the study is very elegant and demonstrates the potential of deep learning approaches in identifying candidate molecules of potential therapeutic use.

Since this reviewer is not an expert on deep learning approaches and pharmacology, I will only focus my review on the biological assays regarding the inhibition of RIPK1. In general, the authors analyze their compound in a necroptosis-assay in vitro and two models of inflammation.

The main weakness in this part of the manuscript is that although the necroptosis pathway seems to be efficiently inhibited in vitro and the compound is effective in blocking TNF-induced systemic pathology and DSS-induced colitis, no direct evidence is presented to demonstrate that this effect is due to RIPK1 inhibition. RIPK1 is a very complex molecule regarding its function in different signaling pathways and its post-translational modifications. The authors also do not present any data regarding the mechanism of RIPK1 functional inhibition. Is this due to blocking one of the phosphorylation sites or the interaction with RIPK3 or scaffolding function independent of phosphorylation?

Specific comments:

The data on TSZ-induced necroptosis comparing RI-962 with the commercial inhibitor GSK3145095 in Figure 7a and b is intriguing. However, an important control for the specificity would be cells in which RIPK1 has been deleted (CRISPR/Cas9) in combination with TNF alone and TSZ stimulation. Figure 7e is not easy to interpret and should be complemented with a quantification of dead cells.

Figure 8 and 9 are also very intriguing regarding the beneficial effects of RI-962. However, the authors have not provided any evidence that this protective effect involves RIPK1. Is this effect mediated by protecting from necroptosis? Is there less activation of RIPK3 and MLKL?

In Figure 8e, the protection of TNF-induced SIRS is not evident in the histologic pictures. Is there protection from TNF-induced necroptosis? From apoptosis?

In Figure 9h and i, the histology suggests that no good care was taken as to where along the colon the samples were collected which is very important when interpreting DSS colitis. The samples showing RI-962 and GSK treated mice clearly show proximal colon. In contrast the DSS-only treated sample shows severe inflammation in the distal colon. As in the DSS-induced colitis model colitis is usually severe in the distal colon while much less activity occurs in the proximal colon, the authors must present samples from the distal colon. If the same samples (from different colon regions) were used in 9e-g, than these analyses also need to be repeated.

Reviewer #3 (Remarks to the Author):

The manuscript by Li and colleagues entitled "Generative Deep Learning Enables the Discovery of a Potent and Selective RIPK1 Inhibitor" describes a computational approach to discover compounds active against RIPK1. Virtual screening carried out against the tailor-made molecular library created by deep learning resulted in eight compounds, which were subsequently synthesized. Encouragingly, four of them showed activities against RIPK1, with one, RI-962, exhibiting an outstanding selectivity for RIPK1. Finally, the potency and selectivity of this compound were investigated by solving its complex structure with RIPK1. This is an interesting study with encouraging results. My only concern is using SMILES in machine learning. This particular representation of chemical compounds was not designed for machine learning applications. Using it with one hot encoding is a poor selection of chemical representation. There are much better approaches designed specifically for machine learning applications, e.g. those converting molecules represented as graphs into embeddings, which are very effective in machine learning.

Reviewer #4 (Remarks to the Author):

I was asked to review the X-ray crystal structure in the paper by Li et.al. I do not know enough about

deep learning to be critical but appreciated the opportunity to learn more about it and the references provide good background for me to develop a better understanding.

Rip1 is a very important drug target with many pharmaceutical company pursuing clinical trials. The authors should expand on the array of compounds in the clinic and the diverse nature of the molecules being pursued. see:

Mifflin, L., Ofengeim, D. & Yuan, J. Receptor-interacting protein kinase 1 (RIPK1) as a therapeutic target. *Nat Rev Drug Discov* 19, 553–571 (2020). <https://doi.org/10.1038/s41573-020-0071-y>

Li, et.al. describe the structure determination of Rip1 bound to RI-962 with clarity and enough detail for one to reproduce the experiment. They are repeating a well established crystal system others have used for Rip1

1. There is a typo in Supplementary Table 5: The R-merge for the high resolution data is (0.0000); this should be corrected.

2. I notice the resolution reported only goes to 2.64Angst but the I/SigI for the high resolution shell is at 2.2. I expect that there is higher resolution data available to refine the model. Why did the authors cut the resolution to 2.64Angst? What is the cc1/2 value? Current standards of x-ray refinement is to extend the diffraction resolution to include reflections with cc1/2 of approx ~0.8. This allows for more reflections to be used and improve refinement.

3. The Rwork/Rfree values seem a bit high. This would be improved with improved refinement. However, the model and conclusions appear to be valid and I would not require reprocessing and refinement. Personally, I would want to submit the best structure model possible, and I think there is room for improvement.

4. I feel the structure description could be improved by comparing RI-962 to the standard RIP1 inhibitor, Nec1, or another more specific Rip1 inhibitor.

5. Does GDL really help find better and novel RIP1 inhibitors compared to what has been done? How does RI-962 compare to the more specific RIP1 inhibitors.

Reviewer #5 (Remarks to the Author):

This manuscript describes a comprehensive study aimed at the development of potent and selective inhibitors of RIPK-1, a Ser/Thr kinase that regulates necroptosis. RIPK-1 is an important therapeutic target, with over 1000 inhibitors developed to date. A known issue of the existing inhibitors is low potency and/or poor selectivity. Hence, there is a need to develop novel scaffolds with improved inhibitory properties. The authors report a new cRNN-based generative deep learning (GDL) model to create a library of compounds that is tailored to RIPK-1. The objective was to overcome problems associated with low chemical diversity of the existing libraries and the often-laborious nature of the fragment-based design approaches. The compounds from the new library were subjected to virtual screening, followed by the synthesis of a selected subset, and evaluation of their potency in biochemical experiments and animal models. Using this approach, a potent and selective RIPK-1 inhibitor, RI-962 was identified.

This is an impressive effort that combines computational and experimental approaches. I have the following concerns regarding the manuscript:

(1) Figures 2c,d require a clear description of different models plotted along the X axis. This description must be included either into the main text or in the Figure 2 legend.

(2) The rationale for showing the entire kinase dendrogram in Figure 5d is unclear, as it does not carry any additional information. The selectivity data as stated in the main text is sufficient.

(3) In the crystallography section, the “DLG-out” conformation is not defined. The allosteric site is not

described in sufficient detail/not clearly identified in Figure 6. The authors report that the activation loop becomes more ordered, but they do not show any data supporting that statement. It is also unclear if there are any existing inhibitors of RIPK-1 that bind to the allosteric site. The binding poses of RI962 and Cpd8 are quite similar (Figure 6c), so the point re unique features of RI-962 ("RI-962 took a novel dual-mode targeting RIPK1") should be expanded upon.

(4) Discussion section: the second half of the discussion section reads like the summary of the work. The discussion would greatly benefit from putting the work into a broader context. For example, how do the authors evaluate the applicability of their GDL model to other kinases? Other biological targets? What are the general limitations of their approach? How large does the target set have to be for the approach to be successful? A thoughtful and thorough discussion would greatly improve the quality of the manuscript.

(5) Minor: there are several instances of typos and/or improper grammar in the manuscript. Some examples are the legend of Figure 2b and the crystallography section of the main text.

Reviewer #6 (Remarks to the Author):

This manuscript reports the discovery of novel highly specific RIPK1 inhibitors by using machine deep learning and AI de novo design approach. I am not an expert in AI drug design area. In term of the medicinal chemistry and biological evaluation parts, the studies was well conducted and the results are interesting and convincing. I support the publication of this manuscript in Nature Comm. There are some issues need to be addressed before its acceptance.

1) Based on their structural features and the binding mode of RI-962 with RIPK1 (co-structure), compounds RI-056 and RI-1155 should demonstrate similar RIPK1 inhibitory potencies or even the target selectivity. It is not clear why the authors did not discuss about these molecules.

2) As the authors mentioned, more than 23 000 new molecules were generated with diverse structural geometries. Are these molecules predicted to bind to the same cavity in RIPK1 with similar poses?

3) In Figure 7f, it is shown that RI-962 equally potent suppresses the phosphorylation of RIPK1 and RIPK3. Whereas, it did exhibit kinase inhibition against RIPK3 in the kinase assay. Any explanation on these results?

Responses to Reviewers' Comments

Referee #1:

“The manuscript entitled "Generative Deep Learning Enables the Discovery of a Potent and Selective RIPK1 Inhibitor" studied a distribution-learning conditional recurrent neural network model to generate tailor-made virtual screening libraries for given biological targets is proposed. The model was then applied to RIPK1. Their contribution is good and the paper.”

Response:

Thank the Reviewer for the positive comment on our work.

Point 1:

“The review of related work is not sufficiently thorough and not sufficiently specific. The authors do not discuss important aspects of the cited works, such as their relative advantages/disadvantages, the results that were reported, or how the different techniques compared to one another and their proposal. Also, although the referenced works are related in terms of the application discussed, this section should be focused on works that employ techniques that are much more similar to the one presented. It should not be difficult to focus on those works that are much more closely related to the methodology presented in the paper and discuss these in-depth.”

Response:

Thanks. We completely agree with the Reviewer. Following the Reviewer's comment, we have discussed important aspects of the cited works, particularly works that employ techniques that are much more similar to our model. Detailed revisions regarding this comment please see pages 3–5.

Point 2:

“The structure of the paper should be added to the end of the introduction.”

Response:

Thanks. This has been done (see page 6, lines 3–12)

Point 3:

“The proposed approach is described quite superficially, with a noticeable absence of references in which the reader could find the details of the different techniques involved. Publication in a top journal such as Natural Communications requires more than just intuitions regarding what a proposed method should achieve: there should be in-depth explanations and formal discussions about how and why the method works, perhaps including experimental evaluation or proofs for each separate module. A comparative discussion of the -detailed- proposal against other similar methods can also be expected. A discussion of the challenges encountered and examples in which the proposed method may fail would also improve the manuscript; in its present form, it would

appear that the formulation and application of the proposed method are straightforward, that there are no variables and no possible conditions that may have been unaccounted for by the authors that could alter its performance.”

Response:

Thank the Reviewer for this comment. Following the Reviewer’s comment, we have added an in-depth description regarding the proposed approach. Detailed revisions regarding this comment please see pages 3–5, pages 6–8, pages 17–19, pages 20–22, Fig. 2–4, Supplementary Fig. 1–5, Supplementary Table 6, Supplementary Notes 1–4, Supplementary Code, and Zenodo software (<https://doi.org/10.5281/zenodo.7074218>).

Point 4:

“The computational details are not mentioned. Which software, program languages, libraries, etc. were used to build this approach? Is it possible to publish code to use this architecture and reproduce the results?”

Response:

Thanks. We have added the computational details in the revised manuscript. We have also complied with the relevant requirements of "Computer Code" by *Nature Communications* and provided necessary supplementary materials for computational details.

(1) "Code and Software Submission Checklist. pdf" and "Supplementary Code.zip" are provided. Further, we also provide a “README” file in our "Supplementary Code.zip" as guidance to run our code.

(2) Our code has been uploaded and published at Zenodo (<https://zenodo.org/>), and the DOI link is provided in the “Code availability” of the manuscript as “Computer code of our GDL model is available as software (source code) at Zenodo (<https://doi.org/10.5281/zenodo.7074218>).”.

(3) The software, the program languages, and the libraries we used to build this approach are provided in the “Data collection” and “Data analysis” of the “Software and code” section in the “Reporting summary.pdf” file, as well as in the corresponding sections of the method part of the main text.

Point 5:

“The authors describe the used methods in detail. However, this is more literature information. It would enhance the quality of the paper, if more information about the implementation and algorithms of the novel workflow would be provided, too.”

Response:

Thank the reviewer for this comment. We have provided more information about the implementation and algorithms of the novel workflow. Detailed revisions regarding this comment please see pages 6–8, Fig. 2a, Supplementary Code, and Zenodo software

(<https://doi.org/10.5281/zenodo.7074218>).

Point 6:

“Lack of coherence, clarity, and poor consistency (e.g., reference is not in the same format).”

Response:

Thank the Reviewer for this comment. We have carefully checked and revised our manuscript including the format of references.

Point 7:

“The motivation part can be elaborated in the abstract, in the introduction as well as in the conclusion, if possible, which will enable the authors to improve the academic merit of their work.”

Response:

Thanks. We have added the motivation part in the abstract, in the introduction as well as in the discussion (conclusion) in the revised manuscript.

Point 8:

“A brief description of hyperparameter optimization (if done), architecture selection (n. of layers), and training time should be provided.”

Response:

Thank the Reviewer for this comment. We have provided a brief description of hyperparameter optimization (see Supplementary Table 6), architecture selection (see Supplementary Table 6), and training time (see README in Supplementary Code).

Point 9:

“Fig 3 quality needs to be improved.”

Response:

Thanks. This has been done.

Point 10:

“In the Method part, only the notations are given. Please state the problem in detail.”

Response:

Thank the Reviewer for this comment. We have provided the detailed description regarding the method we used (see Supplementary Notes 1–4; page 22, lines 11–16; page 25, lines 10–11).

Point 11:

“The proposal is correct and the results are well-founded. There are some fragments of the Discussion that seem more like a summary. I suggest that the Discussion be rewritten to better reflect the quality of the work.”

Response:

Thank the reviewer for this comment. We have rewritten the Discussion section (see pages 17–19).

Point 12:

“The statement and description of future works are needed.”

Response:

Thanks. We have added the statement and description of future works in the revised manuscript (see pages 18–19).

Point 13:

“I suggest to the authors that the manuscript be reviewed by a native speaker to ensure the quality of the writing and the spelling.”

Response:

Thank the reviewer for this comment. We have asked a native English speaker to help us proofread the manuscript.

Referee #2:

“In the current manuscript, Li et al describe an innovative approach of using a distribution-learning conditional recurrent neural network model to generate tailor-made virtual screening libraries for given biological targets and apply this model to RIPK1 in a prove-of-concept approach. Using this approach the authors describe the identification of four compounds that showed activities against RIPK1 one of which was then selected based on superior specificity for RIPK1. The authors show data to demonstrate that their candidate inhibits necroptosis and ameliorates disease in inflammatory mouse models. Overall, the study is very elegant and demonstrates the potential of deep learning approaches in identifying candidate molecules of potential therapeutic use. Since this reviewer is not an expert on deep learning approaches and pharmacology, I will only focus my review on the biological assays regarding the inhibition of RIPK1. In general, the authors analyze their compound in a necroptosis-assay in vitro and two models of inflammation.”

Response:

Thank the reviewer for this encouraging comment and the positive assessment of our study.

Point 1:

“The main weakness in this part of the manuscript is that although the necroptosis pathway seems to be efficiently inhibited in vitro and the compound is effective in blocking TNF-induced systemic pathology and DSS-induced colitis, no direct evidence is presented to demonstrate that this effect

is due to RIPK1 inhibition. RIPK1 is a very complex molecule regarding its function in different signaling pathways and its post-translational modifications. The authors also do not present any data regarding the mechanism of RIPK1 functional inhibition. Is this due to blocking one of the phosphorylation sites or the interaction with RIPK3 or scaffolding function independent of phosphorylation?”

Response:

Thank the Reviewer for this comment. We have added additional experiments to clarify the mechanism of action. The results showed that our compound played its effects by inhibiting the phosphorylation of RIPK1. Detailed revisions regarding this comment please see pages 14–17, Fig. 7g–j, Fig. 8f, g and Fig. 9j.

Point 2:

“The data on TSZ-induced necroptosis comparing RI-962 with the commercial inhibitor GSK3145095 in Figure 7a and b is intriguing. However, an important control for the specificity would be cells in which RIPK1 has been deleted (CRISPR/Cas9) in combination with TNF alone and TSZ stimulation.”

Response:

Thanks. Following the Reviewer’s suggestion, we knocked out RIPK1 in HT29 cells using the CRISPR/Cas9 approach and found that RIPK1 knockout HT29 cells were insensitive to TSZ-induced necroptosis (Fig. 7g, h). Further, western blot showed that RIPK1 knockout cells did not induce phosphorylation of RIPK3 and MLKL under the action of TSZ, which was consistent with the effect of RI-962 (see Fig. 7i, j in the revised manuscript). In addition, induction of TNF α alone (concentrations up to 2 μ g/ml) did not lead to HT29 cell death. All these results demonstrated that RI-962 protects cells from necroptosis by inhibiting RIPK1.

Point 3:

“Figure 7e is not easy to interpret and should be complemented with a quantification of dead cells.”

Response:

Thanks. This has been done in the revised manuscript (see Fig. 7f).

Point 4:

“Figure 8 and 9 are also very intriguing regarding the beneficial effects of RI-962. However, the authors have not provided any evidence that this protective effect involves RIPK1. Is this effect mediated by protecting from necroptosis? Is there less activation of RIPK3 and MLKL?”

Response:

This is an absolutely good suggestion. Following the Reviewer’s suggestion, we explored the mechanism of RI-962 in animal experiments by western blot and immunohistochemistry. Our results showed that RI-962 could significantly inhibit the RIPK1 kinase activity and hence the

phosphorylation of its downstream signaling proteins RIPK3 and MLKL in both animal models, suggesting that the protective effects of RI-962 are mediated by RIPK1 (see Fig. 8f, g and 9j in the revised manuscript).

Point 5:

“In Figure 8e, the protection of TNF-induced SIRS is not evident in the histologic pictures. Is there protection from TNF-induced necroptosis? From apoptosis?”

Response:

Thank the Reviewer for this comment. In Fig. 8e, the protection of TNF-induced SIRS is not evident in the histologic pictures, which could be due to the small magnification of the picture. We reselected the enlarged area and found that TNF-induced SIRS resulted in spotty necrosis in mouse liver and granular degeneration of tubular epithelial cells (arrows in Fig. 8e). Treatment with RI-962 significantly attenuated damage to the liver and kidney by TNF-induced SIRS. A number of studies have shown that RIPK1-RIPK3-mediated cellular damage by necrosis drives mortality during TNF-induced SIRS (Ref: *Immunity* 2011, **35**, 908–918). Meanwhile, our study showed that RI-962 reduced the levels of p-RIPK1, P-RIPK3 and p-MLKL proteins in the liver during TNF α -induced SIRS model, indicating that RI-962 ameliorated TNF-induced SIRS by protecting necroptosis (see Fig. 8f, g in the revised manuscript).

Point 6:

“In Figure 9h and I, the histology suggests that no good care was taken as to where along the colon the samples were collected which is very important when interpreting DSS colitis. The samples showing RI-962 and GSK treated mice clearly show proximal colon. In contrast the DSS-only treated sample shows severe inflammation in the distal colon. As in the DSS-induced colitis model colitis is usually severe in the distal colon while much less activity occurs in the proximal colon, the authors must present samples from the distal colon. If the same samples (from different colon regions) were used in 9e-g, than these analyses also need to be repeated.”

Response:

Thank you very much for pointing out this oversight. It was our mistake that the colon samples were not the same. Therefore, we did the experiment again and provided the sample of the distal colon. Consistent with the original results, RI-962 significantly reduced tissue damage in the colon of DSS-treated mice (see Fig. 9d, e in the revised manuscript). In addition, we used the distal colon samples in both Fig. 9g–i (original Fig. 9e–g).

Referee #3:

“The manuscript by Li and colleagues entitled "Generative Deep Learning Enables the Discovery of a Potent and Selective RIPK1 Inhibitor" describes a computational approach to discover compounds active against RIPK1. Virtual screening carried out against the tailor-made molecular library created by deep learning resulted in eight compounds, which were subsequently synthesized. Encouragingly, four of them showed activities against RIPK1, with one, RI-962, exhibiting an outstanding selectivity for RIPK1. Finally, the potency and selectivity of this compound were investigated by solving its complex structure with RIPK1. This is an interesting study with encouraging results.”

Response:

Thank the reviewer for these encouraging comments and the positive assessment of our study.

Point 1:

“My only concern is using SMILES in machine learning. This particular representation of chemical compounds was not designed for machine learning applications. Using it with one hot encoding is a poor selection of chemical representation. There are much better approaches designed specifically for machine learning applications, e.g. those converting molecules represented as graphs into embeddings, which are very effective in machine learning.”

Response:

Thanks. We acknowledge that there are other molecular representations such as graph-based and fingerprint-based, in addition to the SMILES-based method. Particularly the graph-based method could be a better one to represent chemical compounds in machine learning. Although the SMILES-based method might not be the best one, it has also been used in many investigations to represent chemical compounds in machine learning, and showed good performance (see Refs below). And I do not think that the use of SMILES-based method has a significant impact on our conclusion. But we agree with the Reviewer that SMILES-based method is not the best one. We will in the future adopt more advanced representation methods of chemical compounds in machine learning.

References:

- R1. Atz, K., Grisoni, F. & Schneider, G. Geometric deep learning on molecular representations. *Nat. Mach. Intell.* **3**, 1023–1032 (2021).
- R2. Chuang, K. V., Gunsalus, L. M. & Keiser, M. J. Learning Molecular Representations for Medicinal Chemistry. *J. Med. Chem.* **63**, 8705–8722 (2020).
- R3. Zhavoronkov, A. et al. Deep learning enables rapid identification of potent DDR1 kinase inhibitors. *Nat. Biotechnol.* **37**, 1038–1040 (2019).
- R4. Sousa, T., Correia, J., Pereira, V. & Rocha, M. Generative Deep Learning for Targeted

Referee #4:

“I was asked to review the X-ray crystal structure in the paper by Li et.al. I do not know enough about deep learning to be critical but appreciated the opportunity to learn more about it and the references provide good background for me to develop a better understanding.”

Response:

We greatly appreciate the Reviewer’s positive feedback on the manuscript.

Point 1:

“Rip1 is a very important drug target with many pharmaceutical company pursuing clinical trails. The authors should expand on the array of compounds in the clinic and the diverse nature of the molecules being pursued. see:

Mifflin, L., Ofengeim, D. & Yuan, J. Receptor-interacting protein kinase 1 (RIPK1) as a therapeutic target. Nat Rev Drug Discov 19, 553–571 (2020). <https://doi.org/10.1038/s41573-020-0071-y>.”

Response:

Thank the Reviewer for this comment. Following this comment, we have expanded on the array of compounds in the clinic and the diverse nature of the molecules being pursued. Detailed revisions regarding this comment please see the Introduction section (page 5).

Point 2:

“Li, et.al. describe the structure determination of Rip1 bound to RI-962 with clarity and enough detail for one to reproduce the experiment. They are repeating a well-established crystal system others have used for Rip1.”

Response:

Thank the Reviewer for this positive comment.

Point 3:

“There is a typo in Supplementary Table 5: The R-merge for the high resolution data is (0.0000); this should be corrected.”

Response:

Thanks. This has been corrected in the revised manuscript.

Point 4:

“I notice the resolution reported only goes to 2.64Angst but the I/SigI for the high resolution shell is at 2.2. I expect that there is higher resolution data available to refine the model. Why did the

authors cut the resolution to 2.64 Å?

What is the cc1/2 value? Current standards of x-ray refinement is to extend the diffraction resolution to include reflections with cc1/2 of approx ~0.8. This allows for more reflections to be used and improve refinement.”

Response:

We highly appreciate this constructive suggestion. We tried to cut multiple resolutions for this data set. When the resolution is 2.70 Å, the I/SigI for the highest resolution shell is at 2.8, and the cc1/2 value is 0.819. When the resolution is 2.64 Å, the I/SigI for the highest resolution shell is at 2.2, the cc1/2 value is 0.781 and when the resolution is 2.60 Å, the I/SigI for the highest resolution shell is at 1.8, the cc1/2 value is only 0.617. Normally, the I/SigI for the highest resolution shell no less than 2 is better for refinement. In addition, cc1/2 is about 0.6 (a little bit lower) when the resolution is cut at 2.60 Å. Taking all issues into consideration, we chose 2.64 Å as the final resolution.

Resolution(Å)	I / σ I for the highest resolution shell	cc1/2
2.60	1.8	0.617
2.64	2.2	0.781
2.70	2.8	0.819

Point 5:

“The Rwork/Rfree values seem a bit high. This would be improved with improved refinement. However, the model and conclusions appear to be valid and I would not require reprocessing and refinement. Personally, I would want to submit the best structure model possible, and I think there is room for improvement.”

Response:

Thanks. According to the comment, we have tried to refine this structure with more cycles, but the R_{work}/R_{free} value is still not improved. Admittedly, the R_{work}/R_{free} values seem a bit high. However, we investigated the RIPK1 structures with the similar resolution in the PDB database, the R_{work}/R_{free} values in these structures were close to our results (see below table). We totally agree to submit the best structure model to PDB database.

PDB ID	Resolution(Å)	R _{work} /R _{free}	Refs
6C3E	2.60	0.259/0.307	J. Med. Chem. 2018, 61, 6, 2384–2409
6C4D	2.52	0.236/0.296	J. Med. Chem. 2018, 61, 6, 2384–2409
7YDX	2.64	0.263/0.270	

Point 6:

“I feel the structure description could be improved by comparing RI-962 to the standard RPI1

inhibitor, Nec1, or another more specific Rip1 inhibitor.”

Response:

This is a very good suggestion. Following the reviewer’s comment, we have added the comparison of RI-962 with Nec-1a, in addition to Cpd8, in the revised manuscript. Detailed revisions regarding this comment please see 2.4 section (pages 11–13).

Point 7:

“Does GDL really help find better and novel RIP1 inhibitors compared to what has been done? How does RI-962 compare to the more specific RIP1 inhibitors.”

Response:

Thanks. This is true. By using our GDL model, we have generated more new RIPK1 inhibitors. Just due to the high cost of chemical synthesis, we selected only 8 compounds for synthesis. Regarding RI-962, it is the most potent and selective RIPK1 inhibitor among the 8 compounds we synthesized. RI-962 contains a scaffold, 1-methyl-1*H*-indole-3-carboxamide, which has never been found in other known RIPK1 inhibitors. This compound showed much higher selectivity than other specific RIPK1 inhibitors.

Referee #5:

“This manuscript describes a comprehensive study aimed at the development of potent and selective inhibitors of RIPK-1, a Ser/Thr kinase that regulates necroptosis. RIPK-1 is an important therapeutic target, with over 1000 inhibitors developed to date. A known issue of the existing inhibitors is low potency and/or poor selectivity. Hence, there is a need to develop novel scaffolds with improved inhibitory properties. The authors report a new cRNN-based generative deep learning (GDL) model to create a library of compounds that is tailored to RIPK-1. The objective was to overcome problems associated with low chemical diversity of the existing libraries and the often-laborious nature of the fragment-based design approaches. The compounds from the new library were subjected to virtual screening, followed by the synthesis of a selected subset, and evaluation of their potency in biochemical experiments and animal models. Using this approach, a potent and selective RIPK-1 inhibitor, RI-962 was identified. This is an impressive effort that combines computational and experimental approaches.”

Response:

We thank the reviewer for the positive and kind comment.

Point 1:

“Figures 2c,d require a clear description of different models plotted along the X axis. This description must be included either into the main text or in the Figure 2 legend.”

Response:

Thank the reviewer for this comment. We have added a clear description for the different models plotted along the X axis in the Fig. 2 legend.

Point 2:

“The rationale for showing the entire kinase dendrogram in Figure 5d is unclear, as it does not carry any additional information. The selectivity data as stated in the main text is sufficient.”

Response:

Very good comment. Following the Reviewer’s suggestion, we have removed Fig. 5d.

Point 3:

“In the crystallography section, the “DLG-out” conformation is not defined. The allosteric site is not described in sufficient detail/not clearly identified in Figure 6. The authors report that the activation loop becomes more ordered, but they do not show any data supporting that statement. It is also unclear if there are any existing inhibitors of RIPK-1 that bind to the allosteric site. The binding poses of RI962 and Cpd8 are quite similar (Figure 6c), so the point re unique features of RI-962 (“RI-962 took a novel dual-mode targeting RIPK1”) should be expanded upon.”

Response:

Thank the Reviewer for this comment. Following the Reviewer’s suggestion, we have made revisions.

- 1) We have added the definition of the “DLG-out” conformation (see page 12, line 3);
- 2) We have added more description regarding the allosteric site, and it was also clearly displayed in Fig. 6.
- 3) Regarding the statement that the activation loop becomes more ordered, we think that this is a vague statement, and does not provide valuable information. Therefore, we have removed this statement in the revised manuscript.
- 4) There are RIPK1 inhibitors that bind to the allosteric site, for example Nec-1a, a chemically improved derivative of Nec-1. This has been indicated in the revised manuscript.
- 5) We have rephrased the statement and also added some discussions regarding the binding mode of RI-962 and Cpd8.

Point 4:

“Discussion section: the second half of the discussion section reads like the summary of the work. The discussion would greatly benefit from putting the work into a broader context. For example, how do the authors evaluate the applicability of their GDL model to other kinases? Other biological targets? What are the general limitations of their approach? How large does the target

set have to be for the approach to be successful? A thoughtful and thorough discussion would greatly improve the quality of the manuscript.”

Response:

This is a very good comment. Following the reviewer’s suggestion, we have rewritten the discussion section. Detailed revisions please see pages 17–19 in the revised manuscript.

Point 5:

“Minor: there are several instances of typos and/or improper grammar in the manuscript. Some examples are the legend of Figure 2b and the crystallography section of the main text.”

Response:

Thanks. All these have been corrected.

Referee #6:

“This manuscript reports the discovery of novel highly specific RIPK1 inhibitors by using machine deep learning and AI de novo design approach. I am not an expert in AI drug design area. In term of the medicinal chemistry and biological evaluation parts, the studies was well conducted and the results are interesting and convincing. I support the publication of this manuscript in Nature Comm.”

Response:

Thank the Reviewer for the positive comments.

Point 1:

“Based on their structural features and the binding mode of RI-962 with RIPK1 (co-structure), compounds RI-056 and RI-1155 should demonstrate similar RIPK1 inhibitory potencies or even the target selectivity. It is not clear why the authors did not discuss about these molecules.”

Response:

Thank the Reviewer for this comment. The Reviewer is correct. Compounds RI-056 and RI-1155 indeed demonstrated similar RIPK1 inhibitory potencies with RI-962 (IC₅₀ against RIPK1, RI-056: 0.065 μM; RI-1155: 0.049 μM; RI-962: 0.035 μM). However, both RI-056 and RI-1155 have some cytotoxicity at high concentration (3-10 μM), indicating an off-target effect. This difference between RI-056/RI-1155 and RI-962 might be because compounds RI-056/RI-1155 contain a substructure pyrazolo[1,5-*a*]pyrimidine in their scaffold, which is different from RI-962 (the corresponding one is [1,2,4]triazolo[1,5-*a*]pyridine).

Point 2:

“As the authors mentioned, more than 23 000 new molecules were generated with diverse structural geometries. Are these molecules predicted to bind to the same cavity in RIPK1 with similar poses?”

Response:

Thank the Reviewer for this comment. These 23,000 new molecules were generated by our GDL model, which was trained on the known RIPK1 inhibitors. Therefore, these molecules might be expected to bind to the same cavity in RIPK1 with similar poses. Of course, the real binding poses need to be validated experimentally.

Point 3:

“In Figure 7f, it is shown that Whereas, it did exhibit kinase inhibition against RIPK3 in the kinase assay. Any explanation on these results?”

Response:

Thanks. Our compound (RI-962) is a RIPK1 inhibitor but not a RIPK3 inhibitor. In Fig. 7f, we see that RI-962 inhibits the phosphorylation of RIPK3 because RIPK3 is the direct downstream protein, which means that the RIPK1 inhibition by RI-962 reasonably results in the inhibition of RIPK3 activation.

REVIEWERS' COMMENTS

Reviewer #1 (Remarks to the Author):

The authors have been able to address this challenge at hand with their proposed approach successfully.

Strengths:

- The paper's organization and language are very good and academic.
- The results are important in terms of transferring and fusing knowledge in the related area.
- The new proposed approach to the subject matter is significant in the novelty aspect.

Reviewer #2 (Remarks to the Author):

The manuscript by Li and colleagues has been extensively revised with numerous improvements. My previous comments and concerns have all been addressed.

Two comments remain regarding data quality:

Fig. 7 g,h: The authors should also add the cell viability curve of untreated and TSZ-only treated HT29 cells in both graphs.

I (still) find Fig. 8e and g not convincing. It will be difficult for readers to see a difference. A TUNEL staining and a quantification would help in 8e. I'm also not convinced by the pMLKL staining in 8g. Are these dead or dying cells? why not add pMLKL in the Western blot 8f to have a more quantitative picture?

Reviewer #4 (Remarks to the Author):

I am pleased with the responses to my review

Reviewer #5 (Remarks to the Author):

The authors adequately addressed this reviewer's comments in their resubmission.

Reviewer #6 (Remarks to the Author):

The authors addressed my points properly and I support the publication of this interesting paper.

Responses to Reviewers' Comments

Referee #1:

“The authors have been able to address this challenge at hand with their proposed approach successfully.”

Strengths:

- *The paper's organization and language are very good and academic.*
- *The results are important in terms of transferring and fusing knowledge in the related area.*
- *The new proposed approach to the subject matter is significant in the novelty aspect.”*

Response:

Thank the reviewer for the positive assessment and encouraging comments to our work.

Referee #2:

“The manuscript by Li and colleagues has been extensively revised with numerous improvements. My previous comments and concerns have all been addressed.”

Response:

Thanks.

Point 1:

“Two comments remain regarding data quality:

Fig. 7 g,h: The authors should also add the cell viability curve of untreated and TSZ-only treated HT29 cells in both graphs.”

Response:

Thanks. This has been done in the revised manuscript (see Figure 7g, h).

Point 2:

“I (still) find Fig. 8e and g not convincing. It will be difficult for readers to see a difference. A TUNEL staining and a quantification would help in 8e. I'm also not convinced by the pMLKL staining in 8g. Are these dead or dying cells? why not add pMLKL in the Western blot 8f to have a more quantitative picture?”

Response:

Thank the Reviewer for this comment. We have had a pathologist to help us re-analyze the tissue sections. Now readers can clearly see the difference (see Figure 8e, or the following figure): as indicated by the inflammatory cell infiltration in the portal area of liver tissue, and a glomerular hemorrhage and swelling with neutrophil infiltration in the kidney tissue. Therefore, TUNEL

staining is not necessary. According to the Reviewer's suggestion, we have added pMLKL in the Western blot (see Figure 8f).

Referee #4:

“I am pleased with the responses to my review.”

Response:

Thanks.

Referee #5:

“The authors adequately addressed this reviewer's comments in their resubmission.”

Response:

Thanks.

Referee #6:

“The authors addressed my points properly and I support the publication of this interesting paper.”

Response:

Thanks.